# An optofluidic platform for interrogating chemosensory behavior and brainwide neural representation in larval zebrafish

Samuel K. H. Sy[1,2,3,4,5], Danny C. W. Chan[1,2,6,7], Roy C. H. Chan[1,2], Jing Lyu[1,2], Zhongqi Li [1,2], Kenneth K. Y. Wong [4,5], Chung Hang Jonathan Choi [3,7,8], Vincent C. T. Mok [1,2,9,10], Hei-Ming Lai [1,2,9,10,11], Owen Randlett[12], Yu Hu[13] & Ho Ko [1,2,7,8,9,10,11,14] ✉

Studying chemosensory processing desires precise chemical cue presentation, behavioral response monitoring, and large-scale neuronal activity recording. Here we present Fish-on-Chips, a set of optofluidic tools for highly-controlled chemical delivery while simultaneously imaging behavioral outputs and whole-brain neuronal activities at cellular resolution in larval zebrafish. These include a fluidics-based swimming arena and an integrated microfluidics-light sheet fluorescence microscopy (μfluidics-LSFM) system, both of which utilize laminar fluid flows to achieve spatiotemporally precise chemical cue presentation. To demonstrate the strengths of the platform, we used the navigation arena to reveal binasal input-dependent behavioral strategies that larval zebrafish adopt to evade cadaverine, a death-associated odor. The μfluidics-LSFM system enables sequential presentation of odor stimuli to individual or both nasal cavities separated by only ~100 μm. This allowed us to uncover brainwide neural representations of cadaverine sensing and binasal input summation in the vertebrate model. Fish-on-Chips is readily generalizable and will empower the investigation of neural coding in the chemical senses.

Systems-level investigations of the brain seek to understand the behavioral goals of animals and the underlying algorithmic and neuronal implementations. To this end, sophisticated tools have been developed to mimic the natural habitats and manipulate the sensory environments of an animal for behavioral analysis, as well as to obtain stable, cellular-resolution neuronal activity readouts[1–9]. Larval zebrafish are especially attractive vertebrates for such studies, as they rapidly acquire a rich repertoire of innate chemosensory, auditory, and visually guided behaviors within 1 week after fertilization[10,11], while the optical transparency permits imaging of whole-brain neural and glial activities[7,9,12–16].

Chemosensation is evolutionarily the oldest sensory systems[17]. Considerable efforts have been made to interrogate chemosensory behaviors and the underlying neural basis in several popular model organisms for systems neuroscience (e.g., mouse[1,18–21], zebrafish[22–26], *Caenorhabditis elegans*[27–30], and *Drosophila*[31–35]). Further advancing

the field using larval zebrafish or other small animals requires research tools that can present precisely controlled chemical stimuli to the model system. Typically, in chemosensory behavioral assays, a chemical source is used to maintain the concentration gradient[19,22,23,30,34,35]. In the absence of chemical removal, the overall concentration inevitably rises progressively. In addition, as an animal transverses through the medium, the chemical landscape also varies with time uncontrollably due to molecular diffusion and locomotion-induced turbulent mixing. These changes confound behavioral analyses and the so-obtained results. Studying chemosensory behaviors would greatly benefit from methods that can maintain static chemical profiles in the navigating environment.

In sensory response mapping experiments, in order to dissociate the functional effects of activating individual vs. paired sensory organs, or ipsilateral vs. commissural pathways, a common practice is

to selectively stimulate one or both sensory input channel(s)[24,36–39]. This allows experimenters to reveal how bilateral sensory organ inputs are represented, and the crosstalks between commissural pathways. To record chemosensory neuronal responses, however, chemicals are usually delivered actively to a larval subject in puffs[22,25,26,40] and cannot be restricted to individual nasal cavities. In addition, puffing may introduce confounding mechanical stimulation. Compared to other sensory modalities that do not involve physical contact with the respective cues (e.g., vision[41] and audition[42]), delivering spatiotemporally highly controlled odor cues is especially challenging in larval zebrafish due to their small physical dimensions. For example, while binocular visual interactions can be studied with relative ease by patterned light stimuli[36], the two nasal cavities are only ~100 μm apart and could not be individually stimulated using previously reported experimental platforms for larval zebrafish[43–49] or other small animals[33,35,50–52]. Investigating the neural representations of chemical sensing, therefore also needs a technique that can accomplish the necessary spatial precision for chemical delivery, while minimizing mechanical disturbances, during neuronal activity recordings.

Fluidics and microfluidics (μfluidics)-based techniques offer precise fluid control desired for studying chemosensation, and is thus an attractive approach adopted by numerous previous studies for chemical cue delivery in small animals[44,47,48,53–56]. To additionally tackle the aforementioned challenges, here we report the development of Fish-on-Chips, a set of optofluidic behavioral and neuronal imaging tools for larval zebrafish. Specifically, we use laminar flow settings in two custom-designed fluidic devices for (1) chemosensory behavioral assay in a precisely defined, time-invariant spatio-chemical landscape, and (2) odor stimulus delivery with a few tens micrometer spatial accuracy and fine temporal control, while a custom μfluidics-integrated light sheet fluorescence microscope[14,15,57,58] (LSFM, also known as selective/single plane illumination microscope) is used to perform whole-brain cellular-resolution neuronal activity imaging.

We reasoned that precisely controlled odor stimulus profiles, integrated with behavioral and neuronal imaging, permit unbiased assessment of odorant-evoked behaviors (e.g., regarding valence, behavioral algorithms in pursuit or avoidance), and provide the necessary constraints for constructing neural circuit models among possible alternatives (Supplementary Fig. 1). We hypothesized that this would allow us to address how larval zebrafish avoid potentially noxious chemicals, such as cadaverine —an odor produced by the putrefaction of corpses[59], and whether or how such avoidance depends on binasal inputs. In particular, we will address whether swimming frequency and angular velocity are modulated to assist escape, and if comparisons between the inputs from two sides are used to direct turns. Coupled with functional imaging, we may also uncover the underlying neural basis at the sensory representation level. As a proof-of-principle, we use the navigation assay to reveal the behavioral strategies larval zebrafish adopt to evade cadaverine, and identify a dependence of the avoidance behavior on binasal inputs. Using the μfluidics-LSFM system, we show that the neural representation of cadaverine sensing is characterized by a wide range of ipsilateral-contralateral nasal input selectivity (i.e., from highly ipsilateral or contralateral input-selective, to responding equally to bilateral inputs), and nonlinear summation of bilateral afferent signals on a brainwide scale.

## Results

### A navigation arena with a laminar flow-constrained chemical zone for chemosensory behavioral assay

In the natural environment, a fish needs to alter its swim when encountering fluid streams carrying different attractive or repelling substances. To study chemosensory-guided behaviors under a mimicking condition, we first sought to establish a behavioral assay for larval zebrafish with a precisely defined area where the concentration profile of a chemical remains constant.

We developed a meso- to macro-scale navigation arena with simple geometry and constrained flow patterns—a rectangular arena (6 cm × 3 cm, or ~15 and 7.5 times of typical zebrafish larva body length in length and width, respectively) with three fluid inlets and two fluid outlets (Fig. 1a). In the arena, given the low-Reynolds number ($Re \ll 2000$) flow with Péclet number ($Pe$)>>1 (see Methods), the laminar fluid streams would predominantly slide (or shear) across each layer with minimal lateral mixing. Note that given the design with only 1.5 mm between the ceiling and floor of the arena, most of the flows in the behavioral arena are restricted to 2D, and there is only a very small amount of 3D fluidic mixing. Computational fluid dynamics simulation also suggested laminar flow at steady state with a very low flow velocity over the majority of the area except near inlets and outlets (Fig. 1b). In a chemosensory assay, the rightmost stimulus zone (with several body lengths in dimensions) is filled with a stream of chemical(s) and 3–6 zebrafish larvae freely navigate the arena (Fig. 1a). This assay allows repeated sampling of events of larval zebrafish actively entering and leaving the well-defined chemical zone, which resembles their natural encounters of chemical cues. When instilled with a water stream, the mirror zone serves as a control area with symmetrical flow profile, where the larval subjects' baseline activity in the absence of chemical stimuli can be measured (Fig. 1a). This can also be used to assess the impacts of specific experimental parameters on the swim behaviors (e.g., larvae that underwent different procedures). Additional control assays with water streams filling all zones can be performed in separate animal groups.

By keeping equal flow rates for all three inlets, we were able to maintain over time a clear separation of the fluidic stream zones in the arena in the presence of actively swimming larval subjects. This was verified by infusing an infrared (IR) dye (dissolved in water) into the rightmost zone, and only water into the other two streams, and repeatedly imaging over 2 hours (Fig. 1c). Approximating the concentration profile with IR dye imaging, steep chemical concentration gradients (zero to maximum concentration within ~±1 mm) formed along the chemical zone border between the fluidic stream zones in the arena (Fig. 1d). Upon crossing by a larva, the chemical border is transiently disturbed yet quickly restored in a few seconds (Fig. 1e). We thus verified that the navigation arena (Fig. 1a) can maintain a constant chemical landscape over time, and used it for chemosensory-guided behavioral assays.

### Designing a light sheet fluorescence microscopy-compatible microfluidic chip for precise chemical delivery and stable whole-brain imaging

Apart from behavioral assays, uncovering the circuit principles of chemosensory processing requires highly controlled chemical stimulus delivery with simultaneous neuronal activity recording, preferably by a high-throughput technique such as whole-brain imaging with LSFM[14,15,57,58]. Although agarose embedding has been widely used for the mechanical stabilization of larval zebrafish during imaging[8,12,22,36,37,41,60–64], low-melt agarose gradually dissolves in a fluid medium maintained at ~28° and hinders repeatable chemical stimulus presentation. Moreover, it only allows the delivery of chemicals in puffs with associated mechanical changes, and does not permit spatially restricted odor stimulation (e.g., to the unilateral nasal cavity of a larval zebrafish). Taking advantage of the capability of μfluidics for precise fluid manipulations at a comparable length scale as zebrafish larva body parts, we next aimed to develop a PDMS-based μfluidic device for such purposes. Such a device needs to accomplish several goals, including (i) stabilization of the larval subject, (ii) compatibility with neuronal imaging techniques, and (iii) precise control over chemical stimulus delivery.

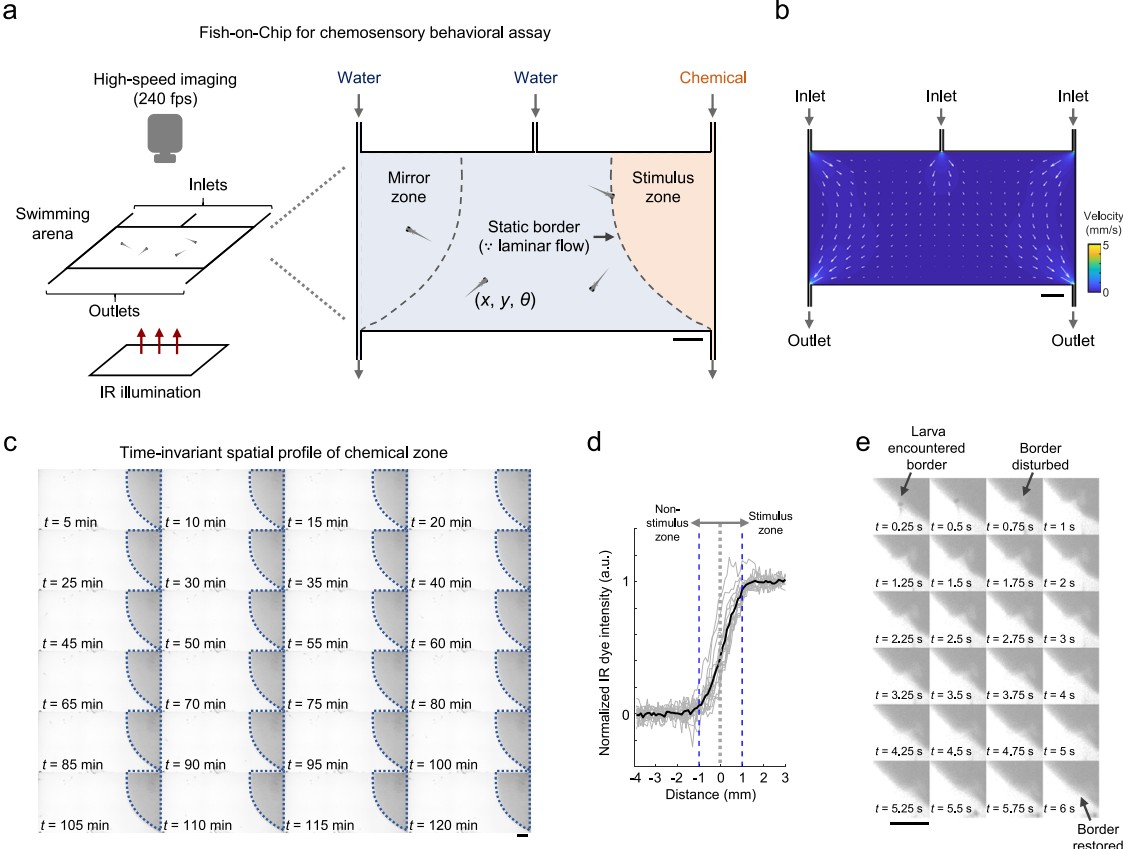

**Fig. 1 | A fluidics-based swimming arena for chemosensory behavioral assay.**
**a** Schematics of the chemosensory behavioral assay. Zebrafish larvae swimming in a two-dimensional arena (60 mm × 30 mm × 1.5 mm) are imaged at high speed (240 fps) under infrared (IR) illumination in the absence of visible light. A chemical zone (stimulus zone) is created and maintained by a constant slow inflow of a given chemical (dissolved in water) via the rightmost fluid inlet. Assays in which all zones are filled by water streams serve as control. The symmetrical mirror zone serves as an additional control for obtaining baseline behaviors when infused by water. The laminar flow maintains a static border between the zones. The coordinates $(x, y)$ and orientation $(\theta)$ of the center of the head are tracked and analyzed for each larval zebrafish. **b** Simulated fluid velocity profile in the swimming arena. Vectors show the direction of flow at the respective locations, with length scaling according to the relative magnitude of velocity. The absolute magnitude of velocity is color-

coded according to the color scale bar. **c** Time-lapsed images (contrast-enhanced) showing the chemical zone border in the presence of larval zebrafish with an IR dye flowing in via the rightmost fluid inlet in a swimming arena identical to that used for chemosensory behavioral assays. Dotted lines in each image outline the same border. Note the subtle differences between the images showing zebrafish larvae navigating the arena. **d** Individual line profiles (gray) of IR dye intensity along normal vectors at different spatial locations of the chemical border (see "Methods") and their mean (black). Positive and negative values on the $x$ axis indicate distances from the border further into and away from the stimulus side, respectively. Blue dashed lines mark ±1 mm from the border. **e** Time-lapsed images (contrast-enhanced) showing transient border disturbance and restoration in an example larval zebrafish border-crossing event. Scale bars in **a**–**c** and **e**: 0.5 cm. Source data are provided as a Source Data file.

We first devised a fluidic channel design for the stabilization of a larval subject's head in the micrometer range for high-quality neuronal activity imaging (Fig. 2a). We made a design with a head and waist-trapping chamber, a tail chamber for its free movement, and a front chamber which consists of a T-junction connecting the inlets and an outlet for the manipulation of fluid streams around each nasal cavity (Fig. 2b). To achieve sufficiently stable trapping, we used a side channel with a fluidic resistance higher than that of the front chamber but much lower than that of the fish trapping chamber (Fig. 2b, lower). Without the side channel, the trapped larva would wobble with changes in fluid flow in the front chamber (Fig. 2b, upper, Supplementary Fig. 2f). The side channel in the final design dissipates the pressure changes during flow changes in the front chamber, and allowed us to sufficiently stabilize a larva in situ (Fig. 2b, lower, Supplementary Fig. 2e).

Next, we ensured that the μfluidic chip is compatible with volumetric neuronal imaging techniques. Since we used a glass coverslip to form the roof of the chip, it is intrinsically compatible with confocal or two-photon microscopes that can directly image the larval zebrafish through the glass roof. However, LSFM is also widely applied to the imaging of small animal brains[8,12,22,65,66] since it can offer higher

volumetric rates. To integrate the μfluidic chip with LSFM, a sidewall that can accommodate the incoming excitation laser is required. Typically, the sidewalls of standard PDMS chips are rough when obtained by cutting through the solidified PDMS with a cutting blade (Fig. 2c)[55,67,68]. We devised a modified fabrication technique that uses a coverslip to obtain a flat transparent sidewall (Fig. 2c), which allows undisrupted propagation of the excitation laser at the air-PDMS interface, and the formation of a light sheet in the μfluidic chip (Fig. 2c).

We then tackled the need for accurate odor stimulation of individual or both olfactory placodes (OPs). This was an unmet challenge due to the small spatial separation of the OPs in larval zebrafish. We noted that the goal could not be achieved by simply delivering parallel fluid streams from the front, since the slight asymmetry and irregularities of the larval subject and/or local flow pressure always led to uncontrolled fluid stream spillover (Fig. 2d). We, therefore, developed a solution with reversed flows from the sides that converge at the midline (Fig. 2e, left and middle). Under laminar flow conditions with $Re << 2000$ and $Pe >> 1$ (see Methods), the lateral streams do not mix and the nasal inputs can sample individual fluid compositions (Fig. 2e, left and middle). Over multiple trials, we invariably obtained a high ratio (>20:1) of sodium fluorescein fluorescence (added to stimulus

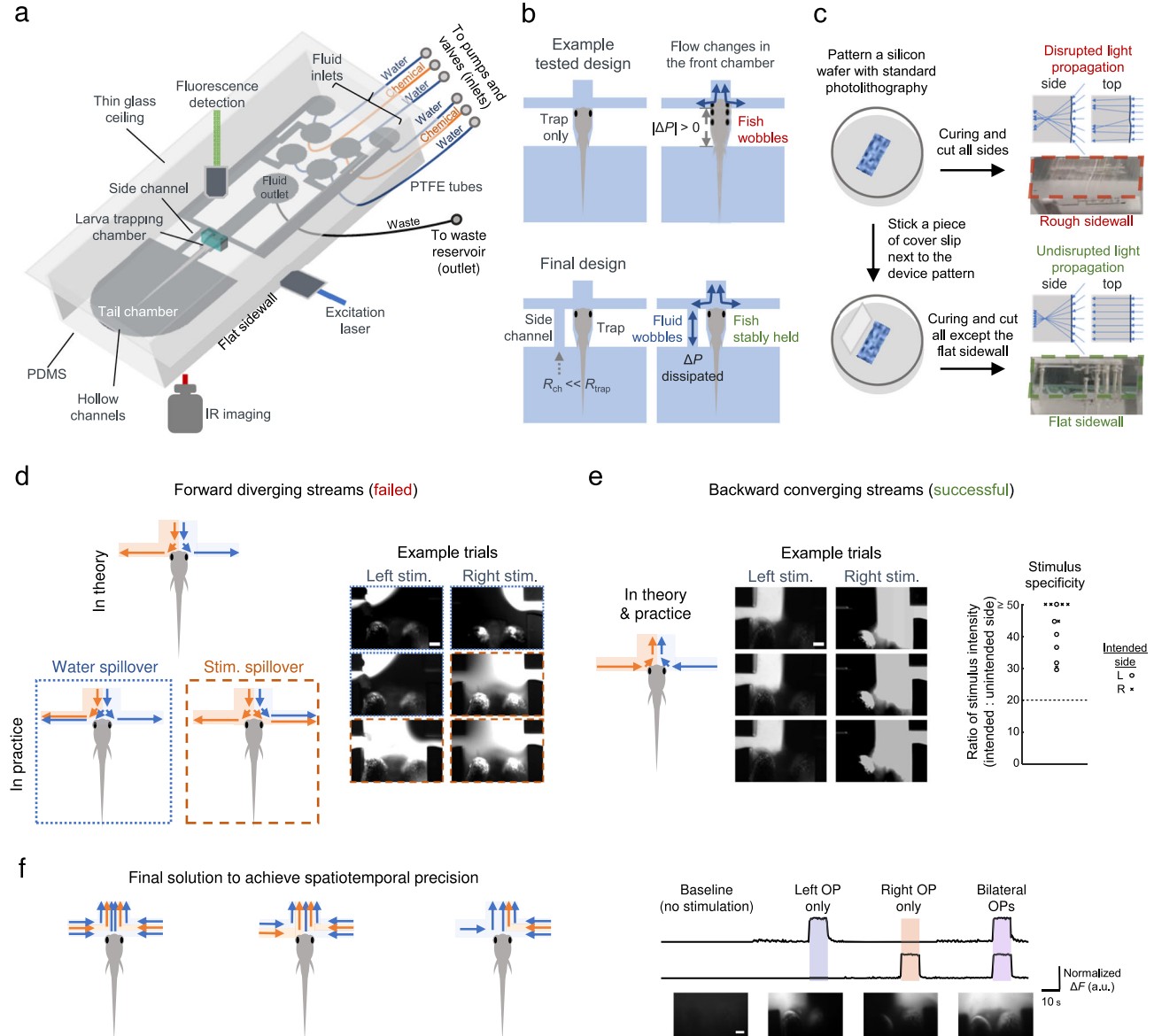

**Fig. 2 | Design principles of the microfluidic components of the optofluidic system. a** Schematics of the PDMS microfluidic (μfluidic) module with a larval chamber, a tail chamber, and a fluid delivery front chamber, which was made compatible with whole-brain and tail imaging. **b** Upper panel: when there are flow changes in the front chamber, changes in pressure difference across the front and tail chambers ($\Delta P$) leads to larva wobbling. Lower panel: with the addition of a side channel that has a much smaller fluid resistance ($R_{ch}$) than that of the trapping chamber ($R_{trap}$), fluid in the side channel dissipates the changes in $\Delta P$ and the larva is stably held. Also see Supplementary Fig. 2e, f. **c** In contrast to rough sidewalls, the flat sidewall ensures undisrupted excitation laser propagation and light sheet formation. The sidewalls are outlined by dashed lines. Schematic side and top views of light ray propagations are shown. **d** Left panel: schematic depictions of the theoretical and actual fluid segregation using forward diverging streams (left-side stimulation illustrated). Blue: water. Orange: chemical (Stim.). Arrows indicate the flow directions. Right panel: example images showing contralateral spillover of chemical or water (specified using corresponding image outlines). **e** Left panel: schematic depictions of the fluid segregation using backward converging streams (left-side stimulation illustrated). Middle panel: example images. Right panel: the ratio of stimulus intensity (intended side vs. unintended side) in test trials ($n = 12$ trials in 2 larvae), quantified with fluorescent imaging using 100 μM fluorescein in the chemical stream. All recorded ratios were >20 (dashed line). Images in **d** and **e** were acquired with three trials for each side with a larva. Scale bars in **d** and **e**: 100 μm. **f** Left panel: the fluidic streams layout that was implemented during different periods of unilateral stimulus delivery (left-side stimulation illustrated). Right panel: visualization and monitoring of nasal stimulation by fluorescence imaging of 1 μM fluorescein in the vicinity of each OPs. Images show one trial for each case with a larva. Scale bar: 100 μm. Source data are provided as a Source Data file.

streams for visualization) in the intended side of stimulus delivery vs. the contralateral side (Fig. 2e, right).

We also optimized fluid stream control to improve the temporal precision of odor stimulation. Flow selection through valve-controlled inlets suffers a relatively long and variable delay of a few seconds for the chemical stream to travel several millimeters through the channels from the inlets to the delivery site. We thus incorporated sandwiched laminar fluid streams on each side, such that the pre-existing chemical

streams only have to travel a ~100-times shorter distance (i.e., ~50 μm only) to reach the nasal cavities amid slow laminar flow (Fig. 2f, left). At the front chamber, valve-gated fluid streams on each side consist of a central chemical stream which is insulated from the nasal cavity by the posterior water stream (Fig. 2f, left). Closure of the valve of the insulating water stream on one side removes the water barrier to the chemical and enables a quick initiation of left or right OP stimulation (l-STIM or r-STIM) (Fig. 2f, left). Unintended stimulation of the

contralateral OP is prevented by the presence of opposing flow on the contralateral side (Fig. 2f, left). Stimulation is turned off by closure of the chemical stream valves (Fig. 2f, left). Bilateral stimulation (b-STIM) is achieved by closure of the insulating water stream valves and terminated by closure of the chemical stream valves on both sides. With a larva in situ, the stimulus profile visualized using sodium fluorescein was characterized by a rapid rise ($0.9 \pm 1.1$ s (mean ± S.D.) to rise from 10% to 90% of maximum values) reaching a plateau (mimicking the encounter of a chemical in a fluid stream), followed by a fast decay phase ($1.6 \pm 1.0$ s (mean ± S.D.) to fall from 90% to 10% of maximum values) (Fig. 2f, right). The difference in rise and decay time arose because it took proportionally more time for 2 streams to refill the space originally occupied by 3 (on stimulus onset), vs. 1 stream to replace 2 (to turn stimulus off) (see Fig. 2f, left).

Note that the overall flow rate slightly decreases with the sequential removal of water and stimulus streams for the initiation and cessation of stimulus delivery, respectively. A continuous flow during baseline ensures minimal shear stress changes throughout stimulus trials, as the spatial profile of relative flow velocity remains the same around the nasal cavities. Moreover, our device also prevents the exposure of any shear stress to the lateral lines. We had thus devised a solution for chemical delivery that not only has a high spatiotemporally precision, but also minimizes the mechanical disturbances intrinsic to other methods that require the initiation of a new fluid stream (e.g., via puffs[22,25,26,40], or forcible ejection of a fluid stream[48]).

## Integrated microfluidics-light sheet fluorescence microscopy system for imaging neuronal activities and behavior

We integrated the μfluidic components with a custom-built LSFM which comprises air, long working distance objective lenses to accommodate the μfluidic setup, and scanning galvo mirror and electrically tunable lens at the excitation and detection arms, respectively (Fig. 3a). In the imaging device chamber, the light sheet formed was 3 mm wide, with the thickness at waist measured from the side = 4.9 ± 0.4 μm (mean full width at half maximum (FWHM) ± S.D.), and the thickness at 150 μm from the waist (i.e., near the edge of the imaging field of view sufficient to cover the majority of a typical larval zebrafish brain) = $6.5 \pm 0.7$ μm (mean ± S.D.). The imaging PSF had a lateral FWHM of $1.2 \pm 0.8$ μm (mean ± S.D.), while the axial PSF FWHM was $2.8 \pm 0.6$ μm (mean ± S.D.). These measurements were close to theoretical values. This μfluidics-LSFM system allowed us to perform volumetric whole-brain cellular-resolution calcium imaging in 5–6 d.p.f. larval zebrafish expressing the calcium reporter GCaMP6f[69] in neurons, spanning 29 planes with 7-μm intervals at 2 Hz volumetric rate (Fig. 3a). Donut-shaped neurons could be seen in most parts of the brain, indicating sufficient stability of larval subject trapping for subcellular-resolution imaging (Fig. 3b). To circumvent the occlusion of the forebrain regions by the right eye, we adopted two imaging configurations with either slight right-tilting (10–20°) or right eye ablation (see Methods for details). Simultaneously, the tail flipping behaviors were monitored using high-speed IR camera imaging at 200 fps (Fig. 3c). Importantly, the optofluidic system permitted stable, repeated imaging of brainwide activity, even during chemical cue delivery to one or both nasal cavities, and when the larval subject's tail flips rigorously (Supplementary Figs. 2 and 3).

In nine behaviorally active larvae (5–6 d.p.f.), the distribution of spontaneous tail flipping frequency (mean ± S.D. = $0.33 \pm 0.32$ Hz) (Fig. 3c, bottom) was comparable to that reported in agarose-embedded larval zebrafish[7]. To image neuronal activities evoked by odor stimuli, a chemical can be presented to unilateral or bilateral OP(s) (Fig. 2). We developed a custom code-based workflow incorporating anatomical registration to the Z-Brain Atlas[70], time-series image registration[71], and calcium signal extraction[72] for integrated data analysis (see Fig. 3d for the workflow, and Fig. 3e, f for the spontaneous calcium activities of an example larval subject, exhibiting higher

baseline activity levels in the olfactory bulbs and the hindbrain). Together with the navigation arena for behavioral recording (Fig. 1), this set of optofluidic Fish-on-Chips tools can be applied to assay chemosensory behaviors and the associated neuronal activities at cellular-resolution evoked by any water-soluble chemical(s).

## Static chemical landscape reveals subtle yet crucial avoidance to cadaverine and its binasal input-dependence in larval zebrafish

We took advantage of the navigation arena of Fish-on-Chips (Fig. 1a) to study larval zebrafish response to cadaverine, an ecologically important diamine product of putrefaction[59]. Although adult zebrafish avoid diamines well[73], previous studies reported that larval zebrafish do not exhibit clear avoidance of cadaverine[25] or some alarm cues[74]. Indeed, by inspecting individual events of encountering cadaverine in the avoidance assay, we also noted that the larvae did not show immediate, strong escape. However, with a longer sampling time (2 hours), our assay revealed that zebrafish larvae unambiguously avoid cadaverine, as they spent substantially less time in the cadaverine zone than the spatially identical stimulus zone filled by water stream in control assays (hereafter referred to as "control") (Fig. 4a, upper). Apart from being an innate aversive cue in adults, we observed that cadaverine is toxic to larval zebrafish, as prolonged compulsory exposure (>60 minutes) to cadaverine (1 mM) in a separate survival assay invariably led to death (Supplementary Fig. 4a). This implied that although such innate avoidance may be subtle and require more observations to reveal, it is crucial to zebrafish larvae.

To further illustrate the applications of the fluidics-based swimming arena, we went on to study the classical problem of bilateral sensory input integration in olfaction, which thus far had been studied at the behavioral level[35,75–79], but the underlying circuit processing rules remained elusive[80]. In theory, bilateral olfactory inputs may serve one or more of the following roles: (1) as physiologically redundant or backup detection channels, (2) enhancing the signal-to-noise ratio of cue detection by summing (or averaging) the two converging signals, and (3) for stereo comparison and spatial information extraction. To determine the necessity of bilateral olfactory inputs in the avoidance behavior, we carried out assays in two additional groups of larvae with two-photon ablation of either the left or the right OP (Supplementary Fig. 4b). In the stimulus zone, although the unilateral OP-intact (uOP) larvae still avoided cadaverine, the proportions of time spent in the cadaverine zone were higher than the bilateral OP-intact (bOP) larvae (i.e., the normal larvae) (Fig. 4a, middle).

Although olfaction is presumably the main cadaverine-sensing modality given the expression of the highly sensitive diamine receptor TAAR13c in zebrafish olfactory sensory neurons (OSNs)[73,81,82], other sensory processes may be involved (e.g., gustation, or chemical-induced malaise). To account for potential non-olfactory component(s) of cadaverine sensing, we carried out assays in an extra larval group with two-photon ablation of both OPs. Interestingly, the null OP (nOP) larvae could still avoid cadaverine, and without further weakening of avoidance compared to uOP larvae (Fig. 4a, left lower). This suggests that cadaverine sensing by larval zebrafish involves non-olfactory detection, and single OP is insufficient to enhance cadaverine avoidance beyond non-olfactory sensing.

By isolating the individual border-crossing events, from entrance to exit of the cadaverine zone, we further quantified avoidance performance with several parameters, including the number of swim bouts (as zebrafish larvae swim in discrete bouts), time and distance traveled to escape for each event. We confirmed that the ablation procedures did not affect baseline swim behavior, since the uOP and nOP larvae exhibited no or minimal differences in these metrics or kinematic parameters from the bOP larvae group in the water-filled mirror zone of the arena (Supplementary Fig. 4c–i). In the stimulus zone, the bOP larvae escaped from cadaverine with fewer bouts than

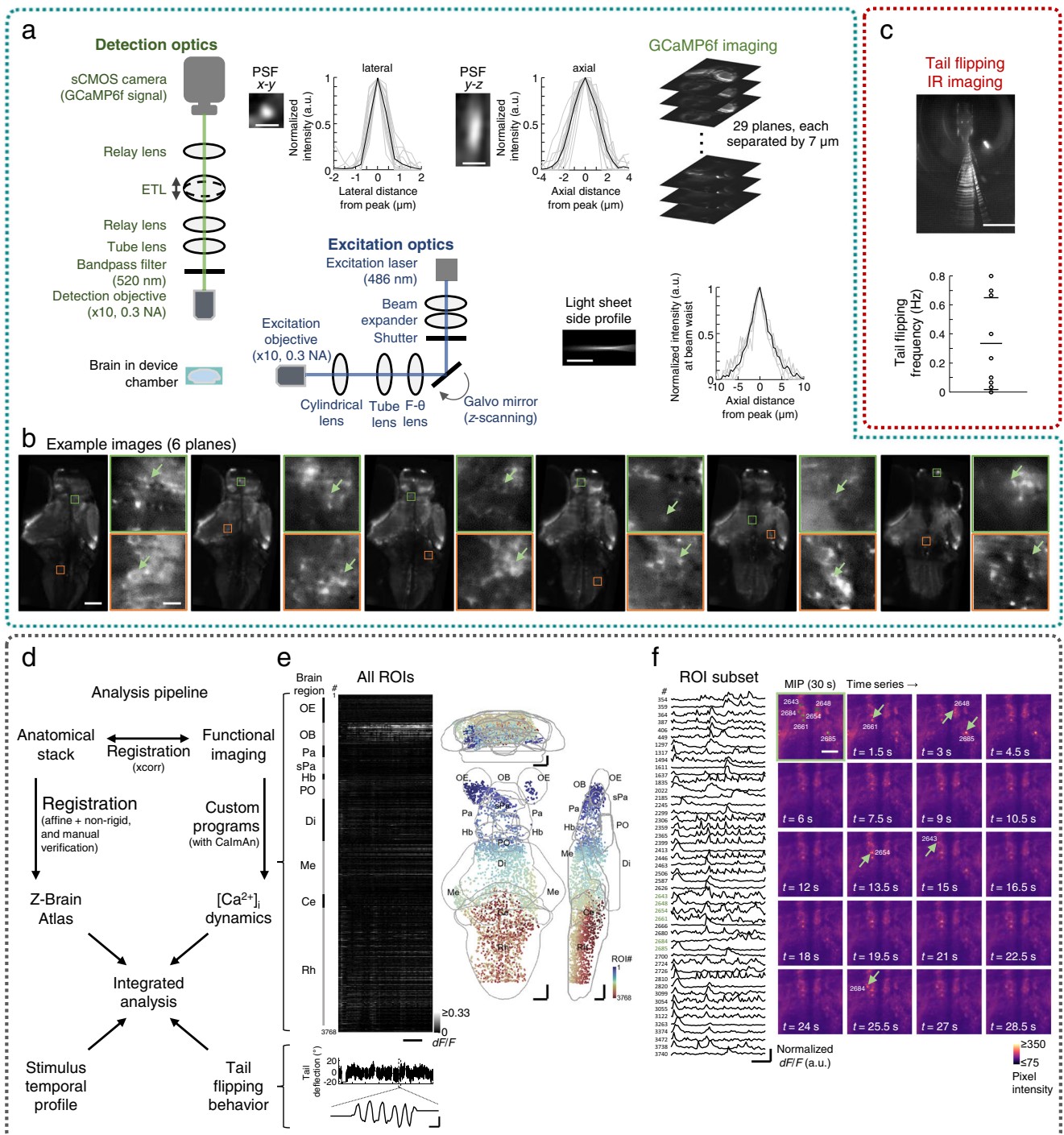

**Fig. 3 | An integrated optofluidic system for in vivo imaging of chemosensory-evoked neuronal activity and behavior. a** The microfluidic (μfluidic) module is integrated with a scanning light sheet microscope for whole-brain imaging in larval zebrafish. Inset image/plot sets show characterization of the excitation and detection arms (see "Methods"). Each set includes an image showing one example profile and a plot of profiles (gray: individual; black: mean), for the lateral (x–y) and axial (y–z) point spread functions (PSFs, scale bars: 2 μm), and the light sheet side profile (scale bar: 50 μm). **b** Example image planes. For each set, the two images on the right are enlarged and contrast-adjusted from the outlined areas of the corresponding larger images. Scale bars: 100 μm for the larger images and 10 μm for the zoomed-in images. Arrows indicate example neurons at various brain locations. **c** Upper panel: temporally overlaid tail images of an example larva (scale bar: 1 mm). Lower panel: spontaneous tail flipping frequency of 9 larvae that were behaviorally active. Horizontal lines indicate mean ±1 SD. **d** Workflow of the custom-developed data analysis pipeline. **e** Brainwide spontaneous activity heatmap and region-of-

interest (ROI) maps (projected to coronal, transverse, and sagittal planes) from an example larva, with 3768 extracted ROIs (each corresponding to a neuron and color-coded by ROI number (#)). Simultaneously acquired tail flipping recording and an example tail flipping event are shown below. Horizontal scale bars for the heatmap: 5 seconds. Scale bars for the event plot: 50 milliseconds (horizontal) and 10° (vertical). Scale bars for the brain maps: 50 μm in Z-Brain atlas space. **f** Left panel: The calcium signal traces of 48 example neurons. Scale bar: 5 seconds. Right panel: a maximum intensity projection (MIP) and the time-series images of the first 30-second interval at a hindbrain region with 6 highlighted ROIs (green: ROI masks). Arrows indicate the locations and times of each ROIs near a calcium event's peak. Brain region abbreviations: OE olfactory epithelium, OB olfactory bulb, Pa pallium, sPa subpallium, Hb habenula, PO preoptic area, Di diencephalon, Me mesencephalon, Ce cerebellum, Rh rhombencephalon. Scale bar for the MIP and time-series images: 50 μm. Source data are provided as a Source Data file.

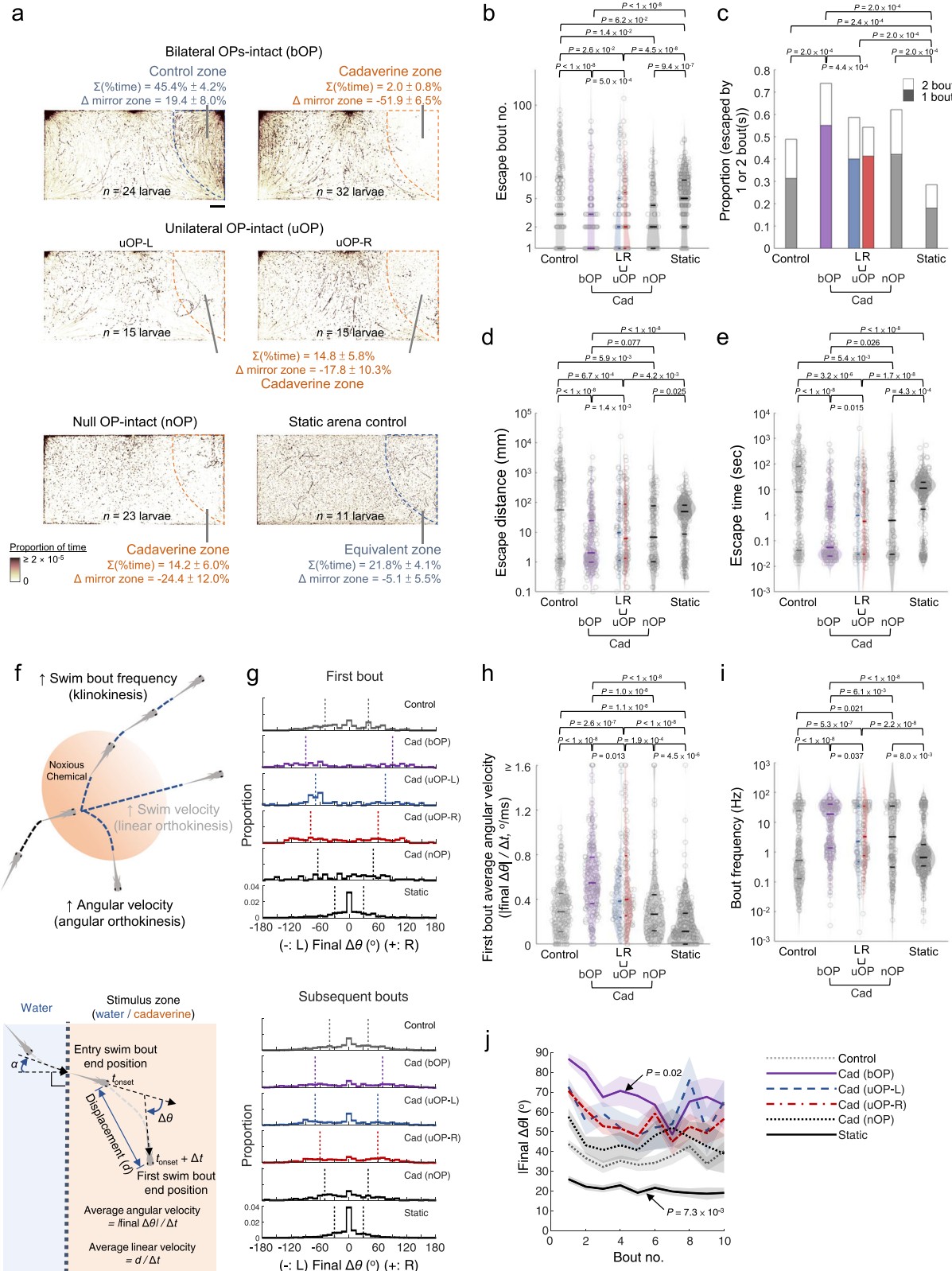

the uOP and nOP larvae (Fig. 4b), and could escape using only one or two bouts in over 70% of the events, while the corresponding proportions for the uOP and nOP larvae in cadaverine zone were only slightly higher than that of control (Fig. 4c). Consequently, the bOP larvae on average traveled shorter distances (Fig. 4d) and took less time (Fig. 4e) to leave the cadaverine zone than the uOP and nOP larvae. In larval zebrafish, the performance of cadaverine avoidance,

therefore, depends on the integrity of bilateral nasal inputs, which do not serve simple redundant roles. There were no differences between the uOP and nOP larvae groups in these avoidance metrics, further supporting that single OPs minimally enhance cadaverine avoidance.

Although constant flows are required to maintain a spatio-temporally invariant chemical profile in the arena that permits an unambiguous analysis of behaviors upon exposure to a chemical of

**Fig. 4 | Larval zebrafish cadaverine avoidance revealed by the chemosensory behavioral assay. a** Upper panels: footprints of bilateral OP-intact (bOP) larvae in control and avoidance assays. Middle panels: footprints of unilateral OP-intact (uOP) larvae in avoidance assays. Lower panels: footprints of null OP-intact (nOP) larvae in avoidance assays and static arena control groups. Note that the static arena control group assays were carried out without flow. For each group, the rightmost (stimulus or water) zones are outlined by dashed lines. The percentages of time spent in the rightmost zone and their differences from that in the mirror water zone (Δ mirror zone) are shown (with SEMs across assays). Scale bar: 0.5 cm. **b** Bout number, **c** proportion of entry-to-exit events with only 1 or 2 bouts (*P* value: one-sided Chi-squared test with Tukey's post-hoc test, comparing 2-bout event proportions), **d** distance traveled, and **e** time taken to escape the rightmost zone in control assays (Control: bOP larvae in the water-only arena with flow; Static: bOP larvae in water-only arena without flow) and avoidance assays (bOP larvae, left OP-intact (L) or right OP-intact (R) uOP larvae, and nOP larvae in arenas with cadaverine

stream in the stimulus zone). **f** Upper panel: illustration of navigational strategies that may be adopted after encountering a noxious chemical. Lower panel: schematics of kinematic parameters that can be extracted. | | denotes absolute value. **g** Histograms of turn angle distributions of first (upper) and subsequent (lower) bouts (i.e., final Δθ). **h** First bout average angular velocity (i.e., |final Δθ|/Δt) upon entering the rightmost zone. **i** Swim bout frequency quantified from all rightmost zone entry-to-escape trajectories. **j** |Final Δθ| vs. bout number (line: mean; shadow: SEM) after rightmost zone entry. *P* value: two-sided Mann–Kendall trend test. **b**, **d**, **e**, **i** The parameters are plotted in log scales. In **b**, **d**, **e**, **h** and **i** Horizontal lines indicate the medians, 75 and 25 percentiles for each group. Shadows of the violin plots scale according to the probability density function. *P* values: Kruskal–Wallis test with Tukey's post hoc test. In **a**–**e** and **g**–**j**, numbers of assays, larvae, and rightmost zone border-crossing events: bOP (control): 6, 24, 211; bOP (avoidance): 8, 32, 251; uOP-L: 4, 15, 71; uOP-R: 4, 15, 155; nOP: 7, 23, 96; bOP Static: 3, 11, 283. Source data are provided as a Source Data file.

interest, the very small flow may still elicit mechanosensory rheotaxis[83,84]. We thus performed an additional set of control assays with bOP larvae in the same arena filled with static water. As expected, the time distribution over different locations was more homogenous, without increased time spent at the fluid inlets (Fig. 4a, right lower) where rheotactic behaviors were observed in the other assays (Fig. 4a). The larvae also took more swim bouts (Fig. 4c), and tended to travel longer distances and times to leave the spatial equivalent zone (to the stimulus zone) (Fig. 4b–e). These however would not confound the comparisons made between bOP, uOP, and nOP larvae in the other assays carried out under identical conditions (i.e., with flowing streams).

### Bilateral olfactory input-dependent modulation of swim bout frequency and turn angle

Next, we sought to understand the behavioral algorithms underlying cadaverine avoidance and further delineate the roles of bilateral olfactory input. In principle, increase in swim bout frequency (i.e., klinokinesis), and angular or linear velocity (i.e., angular or linear orthokinesis) could individually or synergistically account for the observed bilateral OP-dependent efficiency of cadaverine avoidance (Fig. 4f, upper). To distinguish between these possibilities, several swim bout kinematic parameters can be analyzed (Fig. 4f, lower). This also relies on a spatially precise and static chemical landscape, in which the exact concentration of chemical stimulus larval subjects are exposed is known at all locations.

Prompted by the observation that the bOP larvae could escape within two swim bouts more frequently than the uOP and nOP larvae (Fig. 4c), we first focused on the initial swim bouts after encountering cadaverine. Following border-crossing, the bOP larvae made larger turns during the first swim bouts in the cadaverine zone compared to control, with a median angular magnitude difference approximately twice as large as that of the uOP larvae (Fig. 4g). The angular magnitude difference was due to a higher angular velocity of the first bouts by the bOP larvae (Fig. 4h), but not longer bout durations (Supplementary Fig. 5a). With only one nasal input, the uOP larvae were still able to make larger and faster first turns than the control and the nOP larvae, but not as strong as the bOP larvae (Fig. 4g, h). Such an increase in turn velocity and angle, therefore, depends on the nasal inputs in a graded manner. We then analyzed both initial and subsequent swim bouts during the escape journeys. We noted that all larvae groups exhibited higher swim bout frequency in the cadaverine zone compared to the control (Fig. 4i). Increase in swim bout frequency heavily depends on binasal inputs to manifest, as the bOP larvae increased their swim bout frequency substantially more than the uOP and nOP larvae (Fig. 4i). Interestingly, we also observed binasal input-dependent adaptation of swim bout angular magnitude upon continuous cadaverine exposure.

Although both bOP and uOP larvae maintained higher angular magnitudes in the cadaverine zone than control on subsequent swim bouts, the angular magnitude of bouts bOP larvae made converged onto that of uOP larvae with increasing bout number in the cadaverine zone (Fig. 4j).

We did not find the larvae swimming with increased linear velocity or performing directional turns to evade cadaverine. Only minimal differences in swim bout linear velocity in the cadaverine zone was observed between the groups (Supplementary Fig. 5b). If larval zebrafish were capable of stereo comparison and lateralization of odor, the bOP larvae should be able to make directional turns depending on odor zone border-crossing angle, while the uOP larvae should bias their turns towards the ablated side upon encountering noxious odor. However, neither phenomenon was present (Supplementary Fig. 5c, d).

When examining the swimming behaviors of bOP larvae in the static arena, they made less turns and more forward swims than control (Fig. 4g, h, j), whereas the overall swim frequency remained similar (Fig. 4i). These were consistent with the lack of mechanosensory-based rheotaxis. The presence of rheotaxis would not however confound the analysis of chemosensory behaviors in the arena with constant flow, since different experimental groups can be fairly compared under the same conditions.

Collectively, we concluded that under the assay, larval zebrafish combined faster and larger undirected turns and more frequent swim as the innate behavioral algorithm to efficiently escape the noxious cadaverine zone. Although both unilateral and bilateral olfactory detection of cadaverine drive the same behavioral algorithm and that unilateral nasal input can still modulate swim kinematics, binasal inputs are indispensable to optimize the avoidance behavior with higher swim bout frequency and large-magnitude angular velocity increase.

### Revealing brainwide representation of cadaverine sensing using the integrated optofluidic system

The behavioral results suggest similar activation patterns in the larval brain during unilateral or bilateral olfactory cadaverine stimulation, with a more dominant sensory representation in the latter case. Along the early olfactory pathway of larval zebrafish, olfactory bulb (OB) neurons project to bilateral OBs and diverse forebrain regions[85,86], which should mediate bilateral olfactory input integration. However, it is unclear whether the ipsilateral or the contralateral pathway is functionally stronger (Supplementary Fig. 1). Further down the pathways, it is also not known how unilateral or bilateral nasal inputs determine the propagation of evoked neuronal activities. For example, the sites of binasal input crosstalks, and the mechanisms by which bilateral olfactory inputs may enhance sensory representation remained open questions (Supplementary Fig. 1). Empowered by the μfluidics-LFSM system Figs. 2 and 3), we investigated how bilateral

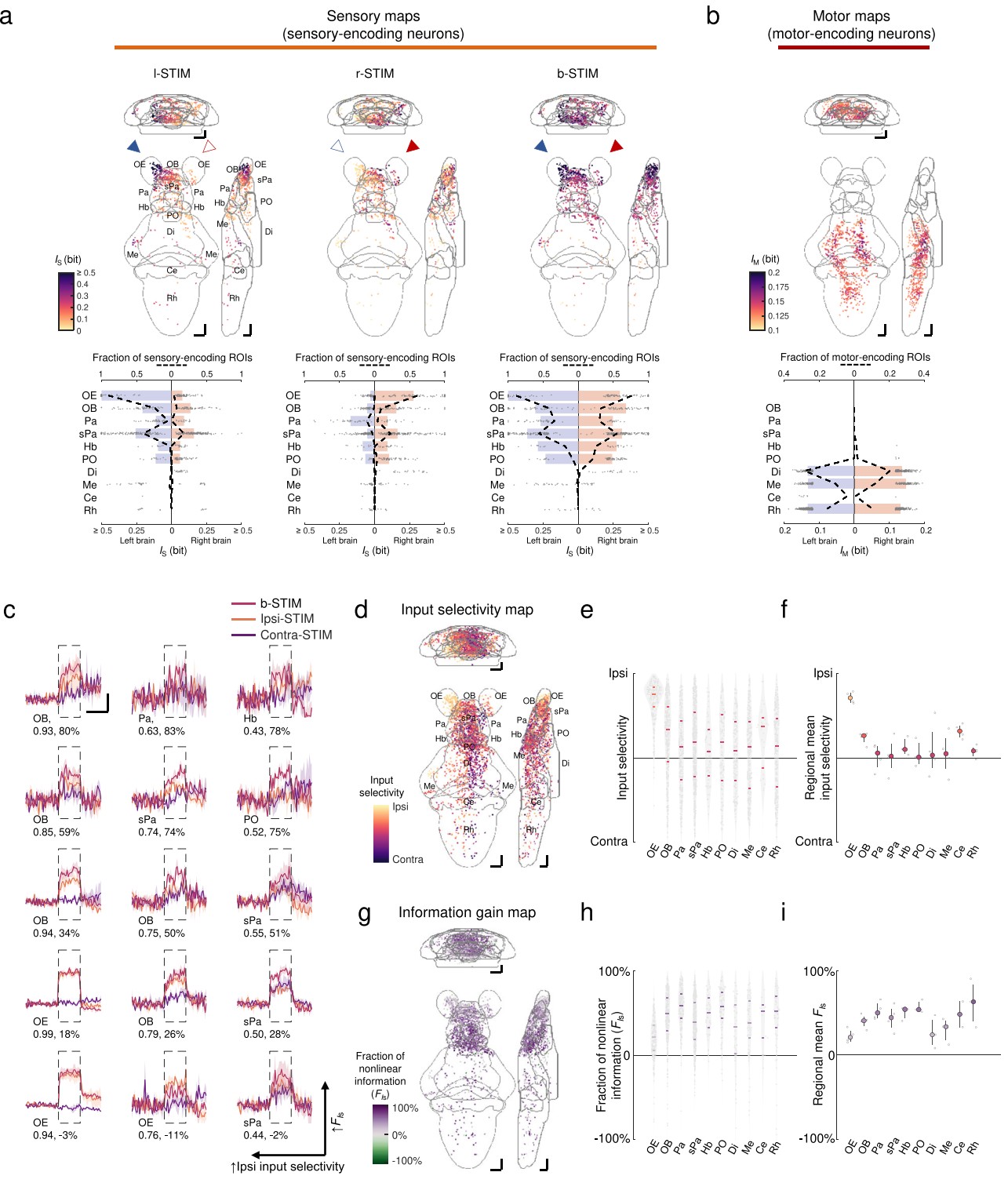

olfactory inputs are processed by downstream circuits to test the hypothesis based on the behavioral results and distinguish between the multiple possibilities (Supplementary Fig. 1).

We identified sensory-encoding neurons, defined as those with the most significant mutual information[87,88] between activity and cadaverine stimulus ($I_S$), to examine whole-brain cadaverine-evoked activity patterns (Fig. 5a, Supplementary Figs. 6a and 7). We used mutual information since it is the most general quantitative measure of dependencies between variables. This also took into account the noisy nature of neuronal activity signals, since its calculation requires the

estimation of the joint probability distribution between stimulus and neuronal activity (i.e., taking the signal variables as statistical in nature). Comparisons of responses using $dF/F$ were also made (Supplementary Fig. 8a). Unilateral OP stimulation results in strong activation of OSNs in the ipsilateral OE (also see Supplementary Figs. 7, 8a–e), confirming the high lateral specificity of stimulation that can be delivered by the μfluidics-LSFM system. Along the olfactory pathway, we found robust bilateral neuronal activation as early as in the OBs. In the forebrain, neurons in the bilateral pallium (Pa) and subpallium (sPa) both respond to unilateral OP stimulation. An asymmetrical

**Fig. 5 | Brainwide neuronal activities evoked by cadaverine stimulation.**
**a** Upper panels: mean intensity projections (to coronal, transverse, and sagittal planes) of the mutual information between the calcium signals of regions-of-interest (ROIs) and stimulus profile of l-STIM (left panel), r-STIM (middle panel) or b-STIM (right panel) ($I_S$), from an example larva. Solid triangles mark the corresponding OP(s) stimulated. Lower panels: corresponding brainwide $I_S$ distributions. Dashed lines indicate sensory-encoding ROI fractions. Bars represent the medians in regions with top six fractions of sensory-encoding ROIs with b-STIM. Abbreviations of brain regions: same as in Fig. 3e. **b** Upper panel: mean intensity projections of mutual information between the calcium signals of ROIs and tail flipping frequency ($I_M$) from the larva in **a**. Lower panel: corresponding brainwide $I_M$ distribution. Dashed lines indicate motor-encoding ROIs fractions. Bars represent the medians in regions with top three fractions of motor-encoding ROIs. **a, b** The numbers of sensory-encoding and motor-encoding ROIs are 676 and 763, respectively. **c** Example trial-averaged responses to ipsilateral (ipsi-STIM, orange), contralateral (contra-STIM, violet), or bilateral (b-STIM, cherry) olfactory stimulation

($n = 3$ trials for each case) of individual ROIs from the designated brain regions with a range of ipsilateral-contralateral input selectivity (first number) and fraction of nonlinear information ($F_{Is}$) (second number). Shadow shows SEMs. Dashed rectangle indicates stimulus window. Scale bars: 10 seconds (horizontal) and 0.5 normalized $dF/F$ (vertical). Data from the same larva shown in **a**. **d** Mean intensity projection maps of ipsilateral(Ipsi)-contralateral(Contra) input selectivity of sensory-encoding ROIs. **e** Brainwide Ipsi-Contra input selectivity distributions of individual ROIs. **f** Regional means of Ipsi-Contra input selectivity. **g** Mean intensity projection maps of $F_{Is}$. **h** Brainwide $F_{Is}$ distributions of individual ROIs. **i** Regional means of $F_{Is}$. **d–i** Data are pooled across larvae ($n = 4$). The number of ROIs in **d** and **e** is 2301, and that in **g** and **h** is 1232. **e, f, h, i** The colors are coded accordingly to the color scale bars in **d** and **g**, respectively. **e, h** Horizontal lines: medians, 75 and 25 percentiles. Shadows of the violin plots scale according to the probability density function. **f, i** Each small dot representing one larva's value. Large dots, upper and lower limits of lines: medians, 75 and 25 percentiles, respectively. Scale bars in **a, b, d, g** 50 μm in Z-Brain atlas space. Source data are provided as a Source Data file.

bulbo-habenular olfactory pathway projects selectively to the right habenula (Hb)[89] which is preferentially activated by cadaverine (at 100 μM)[90] and other odor cues[26,40]. In our dataset, in most larvae the left Hb neurons were similarly activated as the right counterparts by cadaverine (at 1 mM)—a difference that may be due to the higher stimulus concentration we used. The preoptic area (PO), a hypothalamic region known to mediate nocifensive behaviors[61], also generated bilateral neuronal activity during unilateral OP stimulation. Sensory encoding becomes progressively weaker along the rostral-caudal axis in these regions. Beyond the forebrain regions, there is only direct sensory encoding by small subsets of neurons in the diencephalon (Di), mesencephalon (Me), cerebellum (Ce), and rhombencephalon (Rh). Notably, during bilateral OP stimulation, there was symmetrical activation with an increase in overall sensory information content across regions (Fig. 5a, Supplementary Fig. 6a). This is consistent with that predicted by the behavioral results, as the bilateral olfactory input-dependence of cadaverine avoidance implies that some form of input integration for an enhanced representation must be present at the neural activity level.

We then calculated the ipsilateral input selectivity of each neuron (defined as $I_{S\_ipsi-STIM}/(I_{S\_ipsi-STIM} + I_{S\_contra-STIM})$, where $I_{S\_ipsi-STIM}$ and $I_{S\_contra-STIM}$ denote mutual information between calcium activity and ipsilateral or contralateral OP stimulus, respectively). In all regions except the OE, neurons exhibited a wide distribution of input preference, ranging from being highly ipsilaterally selective to highly contralaterally selective (Fig. 5c–f). In the OBs, where subsets of neurons respond to both ipsilateral and contralateral OP cadaverine stimulation, the overall higher ipsilateral selectivity inherited from the OE remains partially conserved (Fig. 5e, f). The ipsilateral input bias becomes weaker in Pa and the sPa, as well as in the other more caudal brain regions (Fig. 5e, f). Similar results were obtained when comparing the $dF/F$ of sensory-evoked calcium transients directly (Supplementary Fig. 8a–d). Therefore, while within each region a wide range of response selectivity is maintained, binasal input signals converge both early in the olfactory processing pathway and in the downstream regions. This indicates a brainwide, distributed mode of parallel input integration.

We also characterized the linearity of bilateral olfactory integration for each sensory-encoding neuron by determining the fraction of nonlinear information (i.e., a fraction of information gained beyond linear summation, denoted by $F_{Is}$). $F_{Is}$ was defined as ($I_{S\_b-STIM}$ - $I_{S\_u-STIM\_sum})/I_{S\_b-STIM}$, where $I_{S\_b-STIM}$ denotes the mutual information of b-STIM-evoked responses and b-STIM stimulus profile, while $I_{S\_u-STIM\_sum}$ denotes that of linearly summed unilaterally evoked responses (see Methods). In principle, a neuron can have sublinear ($F_{Is} < 0$), near-linear ($F_{Is} \approx 0$), or supralinear ($F_{Is} > 0$) gain of information with input summation. Interestingly, apart from observing a supralinear gain of information for the majority of neurons (Fig. 5c, g–i),

there is a trend for $F_{Is}$ to increase along the rostral-caudal axis (Fig. 5h, i) as $I_S$ values decrease across the forebrain regions (Fig. 5a). This was also reflected in the regionally averaged $dF/F$ of calcium signals, as bilateral cadaverine stimulation resulted in the better signal-to-noise ratio of sensory-evoked responses across the different brain regions (Supplementary Fig. 8e, f).

Sensorimotor transformation requires sequential activation of neurons specializing in roles from detecting sensory cues to driving motor outputs. To identify motor-encoding and sensorimotor neurons, we calculated the mutual information between individual neuronal activity and tail-flipping frequency ($I_M$) (see Methods). We defined motor-encoding neurons as those that most strongly encode tail flipping, and sensorimotor neurons as those that at least moderately encode both cadaverine stimulus information and motor output (see Methods). Both motor-encoding and sensorimotor neurons were mostly found in the caudal brain regions where fewer neurons are direct sensory-encoding (i.e., Di, Me, and Rh) (Fig. 5b, Supplementary Fig. 9a, b). The sensorimotor units exhibited supralinear sensory information gain with bilateral olfactory inputs, with $F_{Is}$ higher than their sensory-encoding counterparts in the same regions (Supplementary Fig. 9c). This was also substantiated by examining their regionally averaged responses, as binasal cadaverine stimulation resulted in larger stimulus-locked responses than unilateral stimulation (Supplementary Fig. 9d, e). The enhanced stimulus-locked activities may result in stronger motor unit recruitment, as more motor-encoding neurons were identified when stimulus trials were included than only during baseline trials (Supplementary Fig. 9a, middle and right).

## Discussion

Considerable progress has been made in circuit neuroscience in the past decade. This was in part owing to methodological advancements permitting comprehensive behavioral monitoring and large-scale cellular-resolution imaging of neuronal activities in diverse brain areas[7,10,13–16,91,92]. In this work, we developed optofluidic methodologies for the investigation of chemosensory-mediated behaviors and brain-wide neural representations in larval zebrafish. These included a behavioral assay with a precisely defined and static chemical landscape, and an odor delivery method with high spatiotemporal precisions that are integrated with LSFM for whole-brain imaging.

The fluidics-based swimming arena offers several advantages for assaying chemosensory behaviors. By maintaining a constant chemical landscape, it allows unbiased monitoring of chemosensory behaviors without any assumptions that the behavioral alterations may be strong or subtle. It also facilitates the analysis of specific algorithms underlying chemosensory-guided behavioral changes, by permitting repeated sampling of naturalistic chemical encounters and kinematic parameters while the absolute concentration of chemical exposure is

precisely known. Optimizing the resemblance of chemical sensing to that in the natural environments facilitates the identification of behavioral algorithms adopted under such contexts. We demonstrated these advantages by resolving the apparent discrepancy between the known ecological importance of diamines[59,73] and the reported lack of cadaverine avoidance by larval zebrafish[25]. The fluidics-based assay allowed us to show that the otherwise subtle swim bout frequency and kinematic changes to cadaverine are in fact crucial for mediating and optimizing cadaverine avoidance. This assay can be applied to study responses to other chemicals and natural odorants, and reveal the corresponding behavioral algorithms.

Spatiotemporally precise delivery of sensory stimuli is crucial for mapping neuronal responses. Often, it is important to selectively stimulate one or both of the paired sensory organs, so that the functional interactions of their parallel inputs can be uncovered. This has long been established for studying other sensory modalities in multiple species[36–39]. For the chemical senses, it is comparatively more difficult and had not been achieved in small aquatic animals prior to our study. The primary reason is that the spatial patterns of non-contact-based sensory manipulations are easier to control (e.g., for light), and would not affect the mechanical stability of a larval zebrafish under imaging. In contrast, chemical delivery to a larval subject requires at least one body part being exposed to a fluid medium, introducing challenges in fluid control and mechanical stabilization. Stimulating the unilateral nasal cavity of a behaving larval zebrafish also warrants minimal contralateral crossover or confounding mechanical cues, while any associated changes in fluid flow must not affect the stability of head fixation for optical imaging. Moreover, substantial considerations for integration with optical systems (e.g., the optics-μfluidics interfaces) need to be made when designing the μfluidic components.

To overcome these problems, we developed a method to partition chemical delivery fluid streams at a few tens of micrometer precision, while stably holding the larval subject for cellular-resolution whole-brain imaging with LSFM. With a spatial separation of ~100 μm between the two OPs, the maximal tolerable deviation of midline without contacting the opposite OPs is ~±50 μm, well above the typical length scale of microfluidic flow (~5 μm). We mainly used the laminar property of fluid flow at low-Reynolds number, such that distinct pairs of fluid streams are separately directed to the OPs from the sides and converge at the midline with minimal mixing. To minimize the delay of fluid changes which is intrinsically variable (subject to momentary local fluidic resistance), we aimed to switch fluids as close to the larva as possible. We, therefore, used insulating water streams between the OPs and the cadaverine streams, which can be rapidly switched off for repeated trials of temporally reproducible odor delivery.

Animals are often able to detect chemical concentrations spanning several orders of magnitude due to the high sensitivity of odorant receptors (e.g., TAAR13c for detecting diamines[73,81]). Even small spillovers to the unintended OP (e.g., by fluidic stream spillover, and/or active sampling of water surrounding the nostrils mediated by beating cilia[93,94]), could therefore confound the analysis of neural activity and commissural pathway crosstalks. Nonetheless, the >20:1 lateral specificity our system offers represents the best achievable so far for unilateral odorant stimulus delivery. Importantly, the potential maximal 1/20 spillover would not compromise the interpretations that there is a wide range of ipsilateral-contralateral nasal input selectivity across brain regions (Fig. 5d–f), or the supralinear summation of binasal inputs (Fig. 5g–i), as spillover would only tend to lead to an underestimation of these effects. Aspects of the system still need additional improvements. For example, future works will require improved microfluidic device design that can simultaneously incorporate more fluid inlets and accommodate an additional narrow light sheet to illuminate the forebrain from the front for even more optimal optical access[95].

Collectively, the μfluidics-based behavioral, chemical delivery, and imaging principles we demonstrated with Fish-on-Chips can be readily adapted for studying chemosensory behaviors in other organisms of similar or smaller length scales, such as bacteria, *C. elegans* (see related works[27,56]), *Drosophila* larvae and adults. In nature, an animal would navigate more complex odor landscapes. Yet, the precise spatiotemporal control and homogeneity of odor stimulus in the optofluidic setup allow systematic analyses of the neural representation of individual odorants and evoked behaviors. The high-precision delivery system is also extensible to accommodate an elaborate array of chemicals varying in species, concentrations, and valence, and uncover the corresponding sensory representations of odorants and their mixtures. Additionally, the effects of inter-nasal congruence of odor stimuli (i.e., same vs. distinct chemicals that may also differ in valence) and synchrony of stimulus arrival time can be studied. Other fluid properties (e.g., temperature, viscosity) can also be precisely controlled. Incorporation of crossmodal stimuli will allow the study of their joint effects on the behaviors and neural responses of animal subjects. These will gain us deeper insights into the principles of sensory information processing.

## Methods

### Animals
All experimental procedures were approved in advance by the Animal Research Ethical Committee of the Chinese University of Hong Kong (CUHK) and were carried out in accordance with the Guide for the Care and Use of Laboratory Animals. For all experiments reported in this study, we used 5–7 days post fertilization (d.p.f.) larvae carrying the *nacre* mutation. Larvae were raised and maintained under 14-h light:10-h dark cycles at 28 °C. All larvae used for experiments were not fed (which would not compromise their health[96]). The larvae used for the bilateral OP-intact control, cadaverine, and static arena control groups in the navigation behavioral assay were bred from a pair of heterozygous Tg(elavl3:GCaMP6f) adult zebrafish without screening for GCaMP6f expression. Sex was not determined for zebrafish larvae at this developmental stage. For the unilateral and bilateral OP-ablated groups, the larvae used in the assays had confirmed GCaMP6f expression (by examination under an epifluorescence microscope) to allow structural visualization and two-photon OP ablation.

### Fluidics-based swimming arena for chemosensory behavioral assays
The device for the chemosensory-guided navigation assay was designed in AutoCAD 2018 (Autodesk, USA) and consists of three 1.5 mm-thick laser-engraved poly(methyl methacrylate) (PMMA) layers. The layers were vertically aligned and fused by a chloroform solution applied to the edges of the sheets. This created a closed rectangular arena (60 mm (L) × 30 mm (W) × 1.5 mm (H)) connected to three fluid inlets and two fluid outlets during assays (Fig. 1a). An additional inlet was used for loading larvae and sealed prior to assays. The inlets and outlets are 200 μm wide, prohibiting larvae from exiting the arena through these channels. Syringe needles (Terumo, USA) were sealed at the top layers with epoxy adhesive paste (Devcon, USA) to connect the inlet and outlet channels to fluid-filled syringes and waste collection bottle, respectively, via poly(tetrafluoroethylene) (PTFE) tubes (Cole Parmer, USA). The device was then air-dried for one day to allow evaporation of the residual chloroform and epoxy adhesive paste. The device was rinsed twice with water before use for behavioral assays.

To create a static chemical zone, three fluid streams (two water streams, and one chemical stream with 1 mM cadaverine dissolved in water (pH = 9.0) were formed in the arena by propelling the respective fluids through the inlet channels at 220 μL/min using syringe pumps (LSP02-2A, Longerpump, Beijing) (Fig. 1a). The outlets were connected to open waste collection bottles. The Reynolds number (*Re*) of fluid

flow in the arena is given by $Re = \rho v D_H/\mu$, where $\rho$ is the fluid density, $v$ is the fluid flow velocity, $D_H$ is the hydraulic diameter of the device, and $\mu$ is the fluid dynamic viscosity. Given the rectangular cross-sectional dimensions of the arena (i.e., 1.5 mm × 60 mm), the average volumetric flow rate was 220 μL min$^{-1}$ × 3 inlets = 660 μL min$^{-1}$, which equals 11 mm$^3$ s$^{-1}$. Hence the average flow velocity = 11 mm$^3$ s$^{-1}$/(1.5 mm × 60 mm) = 1.22 × 10$^{-4}$ m s$^{-1}$. $D_H$ for the rectangular channel is given by $2ab/(a+b) = 2.93 × 10^{-3}$ m, where $a$ and $b$ are the dimensions of the rectangular cross-section (i.e., 1.5 mm and 60 mm). Substituting the $D_H$ found, and the density (996 kg m$^{-3}$) and dynamic viscosity (8.32 × 10$^{-4}$ m$^2$ s$^{-1}$) of water at 28 °C, the Reynolds number was found to be $Re = 0.427$. As $Re \ll 2000$, the streams were laminar with negligible mixing[97] and static fluid zone borders were formed (Fig. 1c). The Péclet number ($Pe$) of fluid flow in the arena is given by $Pe = v D_H/D$, where $D$ is the diffusion coefficient of cadaverine (0.66 × 10$^{-9}$ m$^2$ s$^{-1}$)[98]. Substituting the values gives $Pe = 5.42 × 10^2 \gg 1$, cadaverine thus has minimal diffusivity across the streams. Since most small molecules have diffusion coefficients of similar order (i.e., in the 10$^{-9}$ m$^2$ s$^{-1}$ range), the $Re$ and $Pe$ values suggested that chemicals delivered using the system would be minimally crossing via advection and diffusion, respectively, and therefore stay highly localized within individual fluid streams. The static border property of the chemical zone during the entire 2-hour assay period was validated by infusing an IR dye (IR 806, Sigma, USA) into the rightmost stream of the device in the presence of larvae (Fig. 1c). Sharpness of the border was quantified by the relative IR dye intensity (normalized to mean 0 outside, and 1 within the stimulus zone) of 15 line profiles perpendicular to the border evenly distributed from the top to the bottom end (Fig. 1d).

During experiments, the device was leveled and the outflow rates of the two outlets were identical. The two-dimensional (2D) flow velocity profile (Fig. 1b) was obtained by computational fluid dynamics simulation using the CFDTool toolbox (Precision Simulation) in MATLAB (R2018b, MathWorks, USA), assuming incompressible Newtonian fluid flow and with a grid size of 1.4 × 10$^{-4}$ m. The geometry and boundary conditions were set according to the aforementioned device parameters.

### Chemosensory avoidance navigation and survival assays

During a chemosensory avoidance navigation assay, three to six larvae aged 5–7 days d.p.f. were placed in a swimming arena and allowed to acclimate for 15 minutes. Swimming and navigational behaviors were then recorded by a high-speed camera at 240 fps (Mako G-030B PoE, Allied Vision, Germany) and the Streampix 7 (Norpix, Canada) image acquisition software under 850-nm IR illumination (ANGX-1000-CH1-24V, TMS, Malaysia) for 2 hours in the absence of visible light. Fluid temperature was maintained at ~28 °C throughout the assay.

For the control group, all three fluid streams consisted of water. For the static arena control group, the arena is completely filled with water without flow. For the cadaverine avoidance assay groups, the rightmost stream contained 1 mM cadaverine (Sigma, USA) dissolved in water (Fig. 1a), which was freshly prepared prior to each experiment due to the evaporative nature of cadaverine. For the unilateral OP-ablated groups, larvae aged 4–5 d.p.f. were first individually immobilized in 1.5% low-melt agarose (Sigma, USA), then underwent two-photon ablation of OP which was performed under a customized two-photon microscope (Scientifica, UK) with a Ti:sapphire femtosecond laser (Spectra-Physics, USA) tuned to 830 nm. The unilateral and bilateral OP-ablated larvae were released from agarose immediately after ablation and allowed to recover in a water tank for at least 24 hours prior to experiments. Before the behavioral assays, the OPs were embedded in agarose again and imaged under the two-photon microscope to ensure successful ablation (Supplementary Fig. 4b).

For the survival assays, larvae were individually put in a device completely filled with 1 mM cadaverine dissolved in water without flow. Videos of the larvae were recorded at 10 fps, and stimulated with gentle taps to the device every 15 minutes. The assay was terminated when the larva became unresponsive and death was confirmed by the absence of heartbeats and circulation under microscopic examination. No larvae survived continuous cadaverine exposure >165 minutes. A control group of larvae underwent the same assay with devices filled with water for 180 minutes.

### Microfluidic device for precise OP stimulation, behavioral and brainwide imaging

Studying bilateral nasal input integration in larval zebrafish requires fine spatiotemporal control of chemical-carrying fluid flow around the larval subject. On the other hand, stable cellular-resolution neuronal imaging of partially restrained larval zebrafish wholly immersed in continuous fluid medium during OP-specific stimulus presentation must be attained. The PDMS-based microfluidic devices developed to achieve these goals (Fig. 2a) were designed in AutoCAD 2018 (Autodesk, USA). Slight customizations were made to the larva head and waist-trapping chambers to allow the fitting of either a right-tilted or upright-oriented, right-eye-ablated larva. The microfluidic device was fabricated by customized photolithography techniques to incorporate a flat transparent sidewall for excitation light sheet entry, and a thin glass ceiling for emitted fluorescence detection path (Fig. 2c). Briefly, a 2D design was printed on a soda lime mask (Supermask, Shenzhen). Negative photoresist SU-8 2150 (Microchem, USA) was spin-coated on a 4-inch silicon wafer (Semiconductor Wafer, Taiwan) to 500 μm-thick. The device pattern was then transferred to the photoresist with UV exposure (OAI, USA), followed by post-exposure bake and etching to produce a master mold. Mixed PDMS (Dow Corning, USA) and curing agent (at 10:1 w/w) were poured into the salinized mold to a thickness of 5 mm, with a perpendicular glass slide held upright on the mold to produce a flat vertical surface upon curing (Fig. 2c). After solidification at 100 °C for 60 minutes, it was detached from the mold, bonded and sealed onto a microscopic cover slide (as thin glass ceiling) by plasma treatment. The devices were rinsed with water twice before use in experiments.

We derived a fluidic stream control solution to achieve the necessary spatial (Fig. 2e) and temporal precisions of OP stimulation (Fig. 2f, left). On each side anterior to the larva, there were sandwiched laminar fluid streams (water-cadaverine-water) which converged at the midline. The cadaverine stream consisted of 1 mM cadaverine dissolved in water, with 1 μM sodium fluorescein (prepared from a stock solution) for the visualization of chemical delivery in stimulus-response mapping experiments (whereas additional characterization of devices with a larva in situ were carried out with 100 μM sodium fluorescein). Each fluid stream was carried by PTFE tubes, controlled by an solenoid valve (LHDA 0533115H, Lee Company, USA), and driven by a syringe pump. A side channel connecting the chemical delivery microchannels and the tail chamber was incorporated to buffer pressure changes in the fluidic environment, and minimize mechanical disturbances to the larval subject. Flow rates of each stream were maintained at 88 μL/min with slight adjustments for each larva to compensate for the differences in position in the device, ensuring precise convergence of the opposing fluid streams at the midpoint between the OPs. Given the rectangular cross-sectional dimensions of the stimulus delivery front channel (i.e., 0.54 mm (width) × 0.5 mm (height)) (Fig. 2a), the average volumetric flow rate was 88 μL min$^{-1}$ × 6 inlets = 528 μL min$^{-1}$, which equals 8.8 mm$^3$ s$^{-1}$. Hence the average flow velocity = 8.8 mm$^3$ s$^{-1}$ / (0.54 mm × 0.5 mm) = 3.26 × 10$^{-2}$ m s$^{-1}$. $D_H$ for the rectangular channel is given by $2ab/(a+b) = 5.19 × 10^{-4}$ m, where $a$ and $b$ are the dimensions of the rectangular cross-section (i.e., 0.54 mm and 0.5 mm). Substituting the $D_H$ value, and the density and dynamic viscosity of water at 28 °C, the Reynolds number was found to be $Re = 20.3$. As $Re \ll 2000$, the fluid streams would be laminar[97]. The Péclet number ($Pe$) of fluid flow is given by $Pe = v D_H/D = 2.56 × 10^4 \gg 1$, where $D$ is the diffusion coefficient of cadaverine, indicating minimal

diffusivity across the streams. Note that the continuous flow during Pre-stim, stim-on, and stim-off period ensured identical flow profile to the larval subject is exposed to (except for a small overall reduction in flow rate).

## Light sheet fluorescence microscope for whole-brain calcium imaging

We used a custom-built light sheet fluorescence microscope for cellular-resolution whole-brain imaging (Fig. 3a, b). A 486 nm-centered blue gaussian laser (DL-488-050-0, Crystalaser, USA) was used as the excitation light source. The laser beam was resized to 0.6 mm in diameter $(1/e^2)$ by a pair of telescopic lenses (LB1757-A, LB1596-A, Thorlabs, USA), which then passed through a scanning system, followed by a cylindrical lens (LJ1695RM-A, Thorlabs, USA) that focused the horizontal dimension of the parallel beam onto the back focal plane of an air excitation objective (Nikon Plan Fluorite, ×10, N.A. 0.3, 16 mm WD). This expanded the laser horizontally to form a light sheet. The scanning system consisting of a galvanometric mirror (GVS211/M, Thorlabs, USA), a F-theta lens (S4LFT0061/065, Sill Optics, Germany), and a tube lens (TTL200-B, Thorlabs, USA) was used to scan the light sheet vertically and linearly over a range of 210 μm. The F-theta and tube lenses also expanded the beam 3.31 times, resulting in a beam diameter of 2 mm (1/9th of the objective back aperture) and effective excitation N.A. of 0.0332. The excitation light sheet was characterized with 5 measured side profiles along the beam waist (with background pixel intensity subtraction) in micromolar-concentration sodium fluorescein in water, evenly spaced within a 200-μm vertical range.

Along the detection path, a bandpass green filter (525 ± 25 nm) was placed after an air detection objective (Nikon Plan Fluorite, ×10, N.A. 0.3, 16 mm WD) to block the blue excitation light. An electrically tunable lens (EL-10-30-Ci-Vis-LD, Optotune, Switzerland) in between a pair of relay lenses (LA1509-A, Thorlabs, USA) was linearly driven by a lens controller (TR-CL-180, Gardasoft, UK) and synchronized with the light sheet scanner to achieve rapid focusing of different image planes onto the sensor of an sCMOS camera (Zyla 5.5, Andor, UK). Characterization of the point spread function (PSF) was performed by measuring sixteen 0.1-μm fluorescent beads (F8800, Thermo Fisher Scientific, USA), consisting of four samples from each quadrant of the imaging field of view inside the head chamber of the μfluidic device. During an experiment, to correct for axial drift, a reference plane was calibrated with respect to the initial measurement at the beginning of each trial. All control units were synchronized using a multifunctional I/O device (PCIe-6323, National Instruments, USA) and custom code written in MATLAB (R2017b, MathWorks, USA).

## Simultaneous whole-brain calcium imaging and tail-flipping behavior recording

Eighteen 5–6 d.p.f. right eye-ablated and four right-tilted zebrafish larvae underwent simultaneous behavioral and whole-brain calcium imaging experiments, among which nine exhibited spontaneous tail flipping in the μfluidic device (all right eye-ablated), and we obtained stable imaging from nine larvae (six right eye-ablated and three right-tilted). For the right eye-ablated group, 4.5–5 d.p.f. larvae were lightly anesthetized in 0.016% MS-222 and the right eyes were gently removed using surgical forceps under a dissection microscope. Larvae were returned to warmed Ringer's solution and allowed to recover for 12–24 hours before imaging experiments. To minimize mechanical stress in the chamber that the larvae may experience, we only picked suitably sized larvae that fitted the chamber volume well for the whole-brain imaging experiments. For all experiments, larval subjects were only picked if they exhibited normal spontaneous swimming in the tank. Both right before and after imaging experiments, the larvae's heartbeat and response to tactile stimuli were examined. All larvae

included in the datasets exhibited normal heart rate and responded to tactile stimuli.

In an imaging experiment, a single larva was loaded (via the fluid outlet) and fitted into the trapping chamber of the PDMS-based microfluidic device using a syringe pump and monitored under a surgical microscope. The introduction of bubbles to the μfluidic system was carefully minimized during system set up and larva loading, to avoid unwanted fluid flow pattern changes or mechanical instability of the larval subject. The larva was allowed to acclimate for 15 minutes. The axial focus of the light sheet was adjusted to be centered at the brain, correcting for small variations in sidewall width and larva position inside the device. Prior to functional imaging, a detailed anatomical stack of the larval zebrafish brain spanning 138 imaging planes at 2-μm intervals was taken.

Whole-brain calcium imaging spanning 29 planes at 7-μm intervals was performed at 2 Hz volumetric rate, with each trial lasting for 50 or 70 seconds, and ~90-second inter-trial intervals. Trials consisted of 30-second baseline, 10- or 25-second stimulus presentation, and 10- or 15-second post-stimulus epochs. In between the trials, a resting interval was followed by axial drift correction. Each larva underwent 3–4 trials of null stimulus condition recordings and 3–6 trials of each cadaverine stimulus condition (i.e., left, right, or bilateral OP stimulation). During each trial, after a 30-second baseline recording period, the first (water) and the second (cadaverine) stream in front of the OP(s) were switched off sequentially with 10- or 25-second gap to allow the second (cadaverine) followed by the third stream (water) to contact the OP(s), resulting in the initiation and termination of nasal stimulation, respectively (Fig. 2f, left). Recording continued for 10 or 15 seconds after stimulus cessation. Simultaneous tail flipping behavior was imaged at 200 fps under IR illumination by a CCD camera (F032B, Pike, Germany) and the Streampix 7 (Norpix, Canada) image acquisition software.

## Navigational behavior analysis

Individual frames of the navigation behavioral tracking videos were first registered for translation, background-subtracted, and contrast-adjusted. The heading orientation (10° resolution/accuracy) and xy-coordinates (0.1 mm resolution/accuracy) of every larva in each frame were extracted using a semi-automated template matching-based tracking program custom-written in MATLAB (R2018b, MathWorks, US). All tracking results were manually verified on a frame-by-frame basis and corrected when necessary. When larvae came in close proximity (≤ 4 mm) to either the arena boundary or with each other, the trajectories were excluded from analysis of turn behaviors to rule out potential social or mechanical cue-associated movements. A chemical-water border was defined with the aid of an IR dye (IR 806, Sigma, USA) in a separate assay (Fig. 1c). A 1-mm margin along the border was incorporated in analysis, to account for the minimal diffusion and mixing. For each assay, the footprints of larvae in the arena (Fig. 4a) were quantified by summing the head center coordinate occupancy of each pixel by any larva over time and normalized with respect to number of larvae and assay time. The proportions of time spent in the stimulus zone vs. mirror water zone were then quantified by summing values within the respective zones and divided by that over the whole arena. The associated standard errors of mean across assays were also calculated. Larval zebrafish head orientation changes with angular velocity ≥1.2 ° ms$^{-1}$ or linear displacements with velocity ≥12 μm ms$^{-1}$ were registered as swim bouts. The swim bout event detection results were manually verified and corrected when necessary (by S.K.H.S.). Cadaverine zone border-crossing events were identified when swim bout trajectories intersected the cadaverine-water border from the water zone. Subsequent escape trajectories were isolated and analyzed to compare swim bout kinematic parameters and escape efficiency for the different experimental groups.

## Analysis of tail-flipping behavior

Each frame of tail flipping behavioral imaging was background-subtracted and the tail tip-waist angle was extracted by a custom MATLAB program. The tail tip-waist angle was defined as the angle between the long axis of the larval subject body and a straight line joining the tail tip and the waist. The extracted angles were manually verified on a frame-by-frame basis (by S.K.H.S.) and used for the calculations of tail flick event frequency and angular velocity.

## Image processing and calcium signal extraction

Detailed larval zebrafish brain anatomical stacks were registered to the Z-Brain Atlas[70] using affine transformation followed by non-rigid image registration using custom scripts written in MATLAB (R2018b, MathWorks, US). Functional imaging planes were matched to the corresponding anatomical stack by maximization of pixel intensity cross-correlation and manually verified. Stripe artifacts in the anatomical stacks and functional imaging frames were removed using the Variational Stationary Noise Remover (VSNR) algorithm[99]. Functional imaging frames were motion-corrected using the NoRMCorrE algorithm[71]. Trials with blurred frames due to in-frame drifts were discarded. Regions of interest (ROIs) corresponding to individual neurons were then extracted using the CaImAn package[72]. Anatomical landmarks and the regional identity of each ROI were verified by manual inspection (by S.K.H.S.). Note that this only included neurons that had fired at least once during all imaging sessions and with a sufficiently high signal-to-noise ratio determined by the CaImAn-based analysis pipeline.

The stimulus temporal profiles of l-STIM or r-STIM were obtained as the average pixel intensity of a manually defined ROI located immediately anterior to the olfactory epithelium on each side and within the nasal cavities. In trials included for analysis with only unilateral OP stimulation intended, absence of or minimal cadaverine spillover to the contralateral side was confirmed for each experimental trial by intended vs. unintended stimulation side peak fluorescence intensity changes from background ($dF/F$) 20:1 or above. Trials with seizure-like brain activity and/or unsatisfactory image registration were also excluded from analysis. For each larva, 2–4 repetitions or trials per stimulus condition were included for final analysis (except one larva from the group with unbiased response which had only one l-STIM trial).

## Identification of sensory-encoding, motor-encoding, and sensorimotor neurons

The first 10.5 seconds (21 imaging frames) of the imaging data of each trial were excluded from analysis to avoid the inclusion of transient activity evoked by the onset of light sheet illumination. The minimum fluorescence intensity in the subsequent 4.5 seconds (9 imaging frames) of each ROI was used as the baseline fluorescence ($F$). The $dF/F$ signals of remaining imaging frames were calculated as the calcium responses of each ROI. The left ($C_{L,t}$) and right ($C_{R,t}$) cadaverine stimulus profiles were extracted using the abovementioned ROIs directly in front of the respective OPs and normalized to the [0, 1] range. The cadaverine stimulus profiles used in bilateral OP-stimulated trials ($C_{L+R,t}$) were obtained by averaging the left and right stimulus profiles and normalizing them to the [0, 1] range. We calculated the mutual information ($I_S$) between the calcium responses of each ROI and cadaverine stimulus profiles under each stimulus condition by a method based on kernel density estimation of the probability density functions of variables[88]. Each ROI thus had three $I_S$ values, namely $I_{S\_l\text{-STIM}}$, $I_{S\_r\text{-STIM}}$, and $I_{S\_b\text{-STIM}}$, corresponding to $I_S$ values between evoked responses and $C_{L,t}$, $C_{R,t}$, and $C_{L+R,t}$, respectively. To estimate the expected distributions of $I_{S\_l\text{-STIM}}$, $I_{S\_r\text{-STIM}}$, and $I_{S\_b\text{-STIM}}$ due to randomness (shuffled $I_S$), we randomly shuffled the $dF/F$ signals of each ROI (i.e., disrupting their temporal structures) and calculated another set of shuffled values of $I_{S\_l\text{-STIM}}$, $I_{S\_r\text{-STIM}}$, and $I_{S\_b\text{-STIM}}$ (with each ROI contributing one value to each of the distributions). We defined

sensory-encoding neurons to be those that most significantly encode cadaverine stimulus, with at least one $I_S$ value >2.5 times the maximum value of the pooled shuffled $I_{S\_l\text{-STIM}}$, $I_{S\_r\text{-STIM}}$, and $I_{S\_b\text{-STIM}}$ distributions. To calculate normalized $I_S$ (i.e., $I_N$), the $I_S$ of individual ROIs of a given larva were normalized to the mean $I_S$ of its left, right or bilateral OBs, during l-STIM, r-STIM, or b-STIM, respectively.

For each sensory-encoding ROI, we calculated its right input selectivity, defined as $I_{S\_r\text{-STIM}}/(I_{S\_l\text{-STIM}} + I_{S\_r\text{-STIM}})$, where $I_{S\_l\text{-STIM}}$ and $I_{S\_r\text{-STIM}}$ denote mutual information between calcium responses and left or right OP stimulus profiles, respectively. We classified the larval subjects to have biased responses if the mean right input selectivity of all sensory-encoding ROIs (except the OE) deviated from a balanced-response value (i.e., 0.5) by > 0.15 (i.e., > 0.65 or < 0.35). Those with mean right input selectivity ≥0.35 and ≤0.65 were considered generally unbiased. In five stably imaged right eye-ablated and behaviorally responsive larvae, we obtained r-STIM-biased responses in the majority of brain regions (except left olfactory epithelium). This phenomenon was not observed for one right eye-ablated larva and three tilted-imaged larvae included for analysis on nasal input selectivity and information gain (Fig. 5a, Supplementary Fig. 8).

We defined motor output to be the proportion of frames (200 fps) with active tail flipping in each 0.5-s time bins. The calcium response-motor output mutual information ($I_M$) was calculated with the same kernel density estimator-based method using data from all stimulus conditions. Shuffled $I_M$ values were calculated using temporally shuffled motor output traces (with each ROI contributing one value to the distribution). Motor-encoding neurons (Fig. 5b, Supplementary Fig. 9) were defined as those that most significantly encode motor output, with $I_M$ > 2.5 times the maximum values of the shuffled distributions. Motor-encoding neurons were further classified into baseline trial-motor-encoding (i.e., during null stimulus trials) or all trial-motor-encoding units, defined with an $I_M$ during baseline trials >2.5 times the maximum values of the corresponding shuffled distributions, and the complementary set of remaining motor-encoding neurons, respectively.

Sensorimotor neurons (Supplementary Fig. 9) were defined as those with at least moderate encoding of both sensory (at least one $I_S$ > 1.25 times maximum of the shuffled $I_S$ values) and motor output ($I_M$ > 1.25 times maximum of the shuffled $I_M$ values) information. Sensory-only neurons (Supplementary Fig. 9) were classified based on strong sensory encoding (at least one $I_S$ > 2.5 times maximum of the shuffled $I_S$ values) and lack of significant motor encoding ($I_M$ ≤ 1.25 times maximum the shuffled $I_M$ values) (i.e., equivalent to sensory-encoding neurons defined as above with an additional constraint on the lack of motor encoding). Motor-only neurons (Supplementary Fig. 9) were classified based on strong motor encoding ($I_M$ > 2.5 times maximum of the shuffled $I_M$ values) and lack of significant sensory encoding (all $I_S$ ≤ 1.25 times maximum of the shuffled $I_S$ values) (i.e., equivalent to motor-encoding neurons defined as above but with an additional constraint on the lack of sensory encoding).

For the response traces shown (Fig. 5c, Supplementary Figs. 7, 8e, and 9d), normalized $dF/F$ traces were calculated by dividing the $dF/F$ values by the maximum of the trial-averaged traces of all stimulus conditions, to facilitate visualization and comparison of each ROI's or region's averaged responses across stimulus conditions.

## Analysis of input selectivity and fraction of nonlinear information with bilateral input integration

Ipsilateral input selectivity for sensory-encoding neurons was defined as $I_{S\_ipsi\text{-STIM}}/(I_{S\_ipsi\text{-STIM}} + I_{S\_contra\text{-STIM}})$, where $I_{S\_ipsi\text{-STIM}}$ and $I_{S\_contra\text{-STIM}}$ denote mutual information between calcium responses and ipsilateral or contralateral OP stimulus profiles, respectively. For an ROI in the left brain, $I_{S\_ipsi\text{-STIM}}$ and $I_{S\_contra\text{-STIM}}$ were equivalent to $I_{S\_l\text{-STIM}}$ and $I_{S\_r\text{-STIM}}$, respectively. Likewise, for an ROI in the right brain, $I_{S\_ipsi\text{-STIM}}$ and $I_{S\_contra\text{-STIM}}$ were equivalent to $I_{S\_r\text{-STIM}}$ and $I_{S\_l\text{-STIM}}$, respectively.

The fraction of nonlinear information ($F_{fs}$) of each ROI was defined as ($I_{S\_b\text{-}STIM}$ - $I_{S\_u\text{-}STIM\_sum}$)/$I_{S\_b\text{-}STIM}$, where $I_{S\_u\text{-}STIM\_sum}$ denotes the mutual information between linearly summed unilateral OP stimulation-evoked responses and $C_{L+R,t}$ (see above). The linearly summed unilateral OP stimulation-evoked responses were computed by adding the trial-averaged non-preferred OP (i.e., the OP with lower $I_S$ upon stimulation) stimulation-evoked response trace to each preferred OP stimulation-evoked response trace. For this analysis, either ROIs with $I_{S\_b\text{-}STIM} > 2.5$ times the maximum value of shuffled $I_{S\_b\text{-}STIM}$ (Fig. 5g–i) or sensorimotor neurons (Supplementary Fig. 9) were included.

## Statistics

In all plots, only significant or at least marginally significant *P* values (significant level determined at $p < 0.05$) are specified. For multiple larval group comparisons, Kruskal–Wallis test with Tukey's post hoc test was used (Fig. 4b, d, e, h and i, and Supplementary Figs. 4c, e, f, h, i and 5a–c). One-sided Chi-squared test with Tukey's post hoc test was used in comparing proportions between groups (Fig. 4c and Supplementary Fig. 4d). Two-sided Mann–Kendall trend test was used to determine the significance in trends (Fig. 4j). Two-sided Wilcoxon rank-sum test was used to compare two sample sets (Supplementary Fig. 9c). Two-sided Wilcoxon signed-rank test was used to compare two paired sample sets (Supplementary Fig. 9e).

## Reporting summary

Further information on research design is available in the Nature Portfolio Reporting Summary linked to this article.

## Data availability

The AutoCAD files of the devices developed in this study are available at [https://doi.org/10.5281/zenodo.7355156]. Preprocessed data necessary to replicate these results are available at [https://doi.org/10.5281/zenodo.7306139]. Source data are provided with this paper.

## Code availability

The custom MATLAB codes for data analysis are available at [https://github.com/khsamuelsy/FishChip].

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

## Acknowledgements

We thank Tom Mrsic-Flogel and Florian Engert for helpful comments on an earlier version of the manuscript; Jan Schnupp and Vincent Cheung for insightful discussions; Florian Engert for the provision of the Tg(e-lavl3:GCaMP6f) zebrafish. We thank Becky Yung, Florence Yau, Dorothy Ieong, Anki Miu, and Rebecca Chau for administrative support to the project. This work was in part funded by a Croucher Innovation Award (CIA20CU01) from the Croucher Foundation (H.K.); the General Research Fund (14100122) (H.K.), the Collaborative Research Fund (C6027-19GF & C7074-21GF) (H.K.) and the Area of Excellence Scheme (AoE/M-604/16) (H.K.) of the University Grants Committee of Hong Kong; the Excellent Young Scientists Fund (82122001) from the National Natural Science Foundation of China (H.K.); the Margaret K.L. Cheung Research Centre for Parkinsonism Management, Faculty of Medicine, CUHK (H.K.); and the Lo's Family Charity Fund Limited (H.K.).

## Author contributions

S.K.H.S. constructed experimental setups, carried out experiments, and analyzed data. D.C.W.C. contributed to image processing. R.C.H.C. and J.L. assisted imaging experiments. Z.L. assisted in two-photon ablation experiments. H.M.L. contributed to experimental protocol testing. K.K.Y.W., C.H.J.C., V.C.T.M., and O.R. contributed to technical discussions. Y.H. advised on data analysis and interpretation. S.K.H.S., D.C.W.C., and H.K. wrote the manuscript with inputs from O.R. and Y.H., and comments from all other authors.

## Competing interests

The authors declare no competing interests.

## Additional information

[1]Division of Neurology, Department of Medicine and Therapeutics, Faculty of Medicine, The Chinese University of Hong Kong, Shatin, New Territories, Hong Kong SAR, China. [2]Li Ka Shing Institute of Health Sciences, Faculty of Medicine, The Chinese University of Hong Kong, Shatin, New Territories, Hong Kong SAR, China. [3]Department of Biomedical Engineering, Faculty of Engineering, The Chinese University of Hong Kong, Shatin, New Territories, Hong Kong SAR, China. [4]Department of Electrical and Electronic Engineering, Faculty of Engineering, The University of Hong Kong, Pok Fu Lam, Hong Kong Island, Hong Kong SAR, China. [5]Advanced Biomedical Instrumentation Centre, Hong Kong Science Park, Pak Shek Kok, New Territories, Hong Kong

SAR, China. [6]Department of Anaesthesia and Intensive Care, Faculty of Medicine, The Chinese University of Hong Kong, Shatin, New Territories, Hong Kong SAR, China. [7]Peter Hung Pain Research Institute, Faculty of Medicine, The Chinese University of Hong Kong, Shatin, New Territories, Hong Kong SAR, China. [8]Chow Yuk Ho Technology Centre for Innovative Medicine, The Chinese University of Hong Kong, Shatin, New Territories, Hong Kong SAR, China. [9]Margaret K. L. Cheung Research Centre for Management of Parkinsonism, Faculty of Medicine, The Chinese University of Hong Kong, Shatin, New Territories, Hong Kong SAR, China. [10]Gerald Choa Neuroscience Institute, The Chinese University of Hong Kong, Shatin, New Territories, Hong Kong SAR, China. [11]Department of Psychiatry, Faculty of Medicine, The Chinese University of Hong Kong, Shatin, New Territories, Hong Kong SAR, China. [12]Institut national de la santé et de la recherche médicale, Université Claude Bernard Lyon 1, Lyon, France. [13]Department of Mathematics and Division of Life Science, Faculty of Science, Hong Kong University of Science and Technology, Clear Water Bay, New Territories, Hong Kong SAR, China. [14]School of Biomedical Sciences, Faculty of Medicine, The Chinese University of Hong Kong, Shatin, New Territories, Hong Kong SAR, China. ✉e-mail: ho.ko@cuhk.edu.hk

