## [Peer Review File · Nature Communications]

An optofluidic platform for interrogating chemosensory behavior and brainwide neural representation in larval zebrafishREVIEWER COMMENTS

Reviewer #1 (Remarks to the Author):

Sy et al. present an optofluidic platform for studying zebrafish larvae behavior in the presence of chemical cues. The tools include precisely controlled chemical delivery, a microfluidic arena and an integrated light sheet microscopy system. Overall, the authors present an impressive setup with convincing data. The text is very well written.

Major remarks:

- I find it a bit confusing how many different implementations of the microfluidic platform the authors have developed and how they are being used. Is the μ fluidic swimming arena in Fig. 1 actually used for experiments or is this rather a preliminary test on the way to the more complicated setups later on? Fig. 2 and 3 show a very similar setup and it is again not clear if this represents two distinct setups or if these are just two different states during the development of the final system. All in all, this redundancy in the figures should be removed if possible or the authors need to explain in the text the sequence and importance of the different setups.
- Along similar lines, the text is rather lengthy and would benefit from some shortening, making sure it becomes clear what the been why, step-by-step.
- The figures also show some information that is very hard to interpret. For example, in Fig. 1 images in panels c and e look identical. Maybe the contrast can be improved. The photos in Fig. 2d are hard to interpret. It is not clear that the glass window offers any advantage. It is not clear to me what Fig 2g shows. Is it the ratio or what does "vs" mean? Not sure if this is the best way to show these data.
- Figure 3a is confusing as the geometry is not accurate (partially 2D/3D). This needs to be drawn in the correct geometrical perspective.

Reviewer #2 (Remarks to the Author):

The authors have reported the application of two microfluidic devices, one for behavioral screening and one for light sheet brain-wide neuronal GCaMP6f imaging of zebrafish larvae. The former device allows the larvae to move freely in response to cadaverine brought into a chamber at its corners. The latter immobilize the head of the larva while allowing the tail to move in a chamber. The two nasal cavities of the immobilized larva are claimed to be exposed independently to chemicals using laminar flow in the microchannel. The manuscript is well written, and the results are substantial and well presented. There is also a good discussion on the results. I think the manuscript introduction requires substantial revision, and some more experiments are needed to back up some of the claims. If the authors can address these comments, I think the paper becomes suitable for publication in Nature Communications and useful for a wide range of researchers using zebrafish as a model for chemical screening.

Main Critique

One of the most important issues associated with the introduction section of the paper is the lack of a clear research hypothesis, followed by offering no research questions that are investigated. The introduction section is written mostly based on the technologies available to screen zebrafish. On the technology side, zebrafish larvae chemotaxis in chambers and the movement of semi-mobile larvae in microchips have both been demonstrated. LSFM imaging is also well established in the literature. I felt that the prior literature in these areas was not cited thoroughly. The spatial resolution in exposing the nasal cavities independently would be considered a marginal technology improvement. So overall, although very interesting in terms of technology integration and application., the work is not technologically novel enough for the esteemed Nature Communication. From the science perspective, as mentioned above, I failed to see a coherent introduction section with a clear hypothesis and research questions. They are offered in the results section though, and very interesting knowledge gaps have been addressed. So, I recommend re-writing the introduction based on the science

investigated.

Major comments:

- 1- Exposure of the larva's head to the chemical stimulus is accompanied by co-exposure to mechanical shear stress. This shearing flow may also be present in the swimming arena device as there is a continuous flow into and out of the chamber, although being very small. Control experiments measuring the effect of these shears and separating them from chemical sensing phenomena observed in the two devices would be good to elaborate on. Did the authors attempt to minimize these flow rates to minimize the shearing effects?
- 2- On page 4, there are claims that adding the side channel stabilizes the pressure that helps with minimizing the larva wobbling. These claims require experimental proof.
- 3- On page 5, the authors claim that the reverse flow is better than a forward flow for nasal cavity exposure. The provided experimental results are not sufficient to back up this claim. A very small spatial imprecision in larva positioning will hydrodynamically change the flow distribution profile regardless of the flow direction. Small larva movement may also affect the fluid distribution around the nasal cavities that was not discussed in the paper.
- 4- What is the effect of diffusion on the distribution of the chemical at the nose location, potentially causing the chemical to diffuse from the target nasal cavity to the other cavity? This is important to report as the flow velocity is claimed to be very small. Fluorescent images of a dye molecule are not sufficient, as the dye is different from the chemical and the imaging sensitivity may not pick up the diffused dye if fluorescence is under the threshold. The authors can do some analytical calculations with the diffusion coefficient, in addition to the calculation of Peclet and Reynolds numbers.
- 5- LSFM imaging is very interesting in the presented paper. Where the LSFM images in Fig 3b acquired when the tail was moving? How did the movement affect the LSFM images?
- 6- I recommend adding a section and fully characterizing the specifications of the light sheet.

Minor comments:

- 1- Page 2, paragraph 2, line 2-3: The sentence reads as if no tools are available for precise chemical stimulation of small organisms. This is an incorrect message and should be revised to acknowledge the substantial work done in this area on *C. elegans*, *Drosophila* and zebrafish using microfluidics.
- 2- Please justify in the introduction section why it is important to expose nasal cavities to chemicals one at a time.
- 3- On page 5, the rise time average is much shorter than the decay time average in chemical stimulation. Can the authors discuss why? Also, how do this difference and large variation in stimulation time affect the experiments claimed to be highly controllable?

Reviewer #3 (Remarks to the Author):

The authors present a technically demanding and impressive study to analyse neural circuits activated by odors, together with the behavior elicited by these odors. Larval zebrafish are used as model system due to their transparency and small size, which allows whole brain imaging. Tail flipping movements can be recorded in parallel with imaging, unfettered movements are recorded in a separate arena with unusually high spatiotemporal control of odor stimuli (on a tens of μM scale). The main (and important) improvement beyond previous such studies concerns the precise control over odor concentration and extension of the odor plume in both settings. The accuracy of the imaging setup is sufficient to stimulate the left and right nostril of the zebrafish larvae separately, thus allowing to demonstrate interhemispheric connections.

The authors have used the aversive odor cadaverine to demonstrate the feasibility of their setup. Adult zebrafish were known to react with aversive behavior to cadaverine, but the response of larvae was not clear from previous experiments. Moreover the authors have imaged the brain networks activated by cadaverine with uni-nasal and bi-nasal odor stimulus, and have compared the behavior of larvae with one olfactory placode removed (to achieve uni-nasal input) with untreated control larvae. The evaluation of the data appears to be thorough.

Reported biological results are:

1. Zebrafish larvae do strongly avoid cadaverine (but see below)
2. Larvae with only one olfactory placode need more movements to leave the cadaverine zone. This is interpreted as binasal input being required for proper motion control.
3. Stimulus-induced activity occurs on the contralateral side already in the olfactory bulb, and all more posterior brain regions.
4. Stimulus-induced activity for animals with binasal input is supralinear compared to stimulus presented only to one nose.

Taken together this is a technically very strong paper, with novel findings enabled by the improved methods, but I have some doubts concerning parts of the biology (see below).

Major points:

- p6: The authors use 1 mM cadaverine for the avoidance assays and state that no immediate effect was seen, but over 2 hours there was strong avoidance. The authors also show that after 2 hours in 1 mM cadaverine the large majority of larvae is already dead. Thus it is unclear, whether the avoidance results from olfactory sensing of cadaverine – it could also come from sensing malaise (a standard method to strongly induce taste aversion is based on such sensing). This could be examined by bilateral ablation of the olfactory placodes. If the author's interpretation is correct, the avoidance behavior should no longer occur. Such control experiments should be included.
- The authors seem to have used a single and rather high cadaverine concentration, i.e. 1 mM. Why was no lower concentration used? Ref 75 shows imaging results with 100 μ M cadaverine. Using 100 μ M could have avoided the problem with mortality (see above). Furthermore, with the 20:1 specificity for unilateral odor presentation reported by the authors (p.5), 1 mM stimulus still means 50 μ M on the contralateral side, which could be enough for a response. This problem would be much reduced with 100 μ M starting concentration.
- p13: The authors state in Methods that they used 5-7 dpf zebrafish larvae. 5 dpf larvae presumably have not yet fed, but 6 and 7 dpf have taken up food. Can the authors exclude that feeding experience does influence the observed behaviors? How was the age distribution for the individual experiments?
- p14: The larvae with unilateral ablation of olfactory placode (uOP) are treated rather differently from the control animals (bOP). I could not find any description of a mock procedure for the bOP, to attempt to control for the stress induced in the uOP (by extra embedding on 4-5 dpf and 2 photon ablation, plus additional 2 photon analysis before the experiment). Can the authors exclude that these animals were in generally poorer shape and therefore show hampered motion?
- A general point, which could be touched upon in the discussion: The strength of the paper is the precise control and temporal and spatial homogeneity of the odor stimulus. However, this is also a fairly unnatural situation, as in nature the animal would navigate a 'textured' plume of odor.

Minor comments:

p2, „As ecologically the oldest sensory systems¹⁷, our understanding of the array of chemosensory-guided behaviors and the neural basis of chemical senses remains incomplete.“

Sentence logic incorrect, please rephrase. Also I suspect 'evolutionary' is meant, not 'ecological'?

p3, „we show that the neural representation of cadaverine sensing is characterized by a wide range of ipsilateral-contralateral nasal input selectivity“

Unclear. What is a 'wide range of nasal input selectivity'?

p3, „the laminar fluid streams would predominantly shear across each layer,,

What does 'shear' mean in this context?

p3, „Computational fluid dynamics simulation showed laminar flow at steady state“
'Suggest' would be a better term than 'show', since it is a simulation.

p5, „(1.64 ± 1.04 s (mean ± S.D.)“

This should at most be given as '1.6 +/- 1.0', the second digit does not carry information.

Figure 3B, p6 top

The authors claim „Donut-shaped neurons could be seen in most parts of the brain“

It would be preferable, if the authors point out some neurons by suitable symbols.

It would also be helpful, if the authors could give a (rough) estimate about the frequency of labeled neurons, e.g. per total cells or per area.

p6, „In 9 behaviorally active larvae, the distribution of spontaneous tail flipping frequency „

The age of these larvae is not given. Since the behavior of young larvae depends very much on their age, this is an important information, which should be given.

p8, „We did not find the larvae performing significant linear orthokinesis „

The paper would be easier to read for the non-specialist if these and other technical terms would be replaced by their explanations (given on p7) – i.e. 'linear velocity' for 'linear orthokinesis' and 'swim bout frequency' for 'klinokinesis'.

p8, „We identified sensory-encoding neurons, defined as those with the most significant mutual information^{72,73} between activity and cadaverine stimulus „

Please explain 'mutual information'. How does it account for 'noisy baseline'?

p9, „In our dataset, in most larvae the left Hb neurons were similarly activated as the right counterparts by cadaverine (at 1 mM).“

Maybe speculate a bit why this result is different as that obtained in ref. 75.

p9, „there is an increasing trend of FIs (Figure 5g, h) as IS values decrease „

Do the authors mean that there is a trend of FIs to increase?

p10, bottom, „reported lack of cadaverine avoidance by larval zebrafish^{29,63}.“

Ref. 63 is a 1982 paper about the alarm response of zebrafish. Cadaverine is not mentioned in this paper.

p15, „Slight customizations were made to the larva head and waist-trapping chambers to allow the fitting of either a right-tilted or upright oriented, right eye-ablated larva.“

Please explain the mechanical stress (or lack thereof) for the larvae in these chambers.

Rebuttal Letter

Summary of Revision

We thank the reviewers for the overall positive evaluations and encouraging remarks, as well as all the suggestions and constructive criticisms. We have performed additional experiments and analyses, and revised the manuscript extensively based on the reviewers' comments, and at places where we felt more clarity was needed. Major additions and changes include (all changed texts and figure panels are highlighted in red and underlined in the main text or corresponding figure legends):

- In the revised Introduction, we now provide better coverage for the background literature (**page 2, paragraph 2 & page 3, paragraph 1**). We also improved the elaboration on the importance of our technical advancement on attaining a spatial accuracy sufficient to specifically stimulate unilateral olfactory epithelium (**page 2, paragraph 3**), and outlined a clear hypothesis for the subsequent behavioral and neural investigations on the bilateral olfactory input dependence of cadaverine avoidance in larval zebrafish (**page 3, paragraph 2**).
- Regarding the general presentation of the design principles and technical advantages of the microfluidic (μ fluidic) device for whole-brain imaging: We rearranged **Figures 2 & 3** to remove redundant information, added more schematics and data to better showcase the advantages of the final design on different aspects (**Figure 2c–e, Supplementary Figure 2e, f**). We also revised the corresponding main text to better explain the stepwise design goals (**page 4, paragraph 3 to page 6, paragraph 1**), so that the logical deduction and steps of how we derived the final design is more clearly presented.
- We added physical characterizations for the optical imaging parameters of our integrated μ fluidics–light sheet fluorescence microscopy system (**Figure 3a**). We also provide additional data and images to show that the system allows the experimenter to perform unilateral nasal placode-specific chemical cue delivery (**Figure 2d, e**), while simultaneously carrying out stable, cellular resolution neuronal and behavioral imaging even during rigorous tail movements (**Supplementary Figures 2 & 3**).
- We carried out additional numerical analyses to address the concerns on diffusive mixing in both the behavioral and imaging setups. The large Péclet number ($\gg 1$) and small Reynolds numbers ($\ll 2000$) (i.e., in reasonable orders of magnitudes away from their respective thresholds) further substantiated our claim that when using our μ fluidic devices, turbulent and diffusive mixings would minimally affect the desired segregation of cadaverine (or other small-molecule chemical cues) from water streams (behavioral arena: **page 4, paragraph 1**; μ fluidic device for whole-brain imaging: **page 5, paragraph 4**; see detailed calculations in **Methods, page 16, paragraph 1 & page 18, paragraph 1**).
- We performed new behavioral assays in larval zebrafish with both olfactory placodes (OPs) ablated (designated null OP-intact (nOP) larvae, $n = 23$ larvae, with 3 – 6 larvae per assay in 7 experiments) (**revised Figure 4**). These experiments strengthened our results on the roles of binasal inputs in cadaverine avoidance, as we found that bilateral OPs are required to fully drive angular speed increase and swim bout frequency increase.

In contrast, a single nasal input can only partially mediate angular speed increase, and fails to enhance swim bout frequency or cadaverine avoidance beyond that without any intact OPs.

- To address the effects of shear stress exposure on the larval subjects: We performed new control behavioral assays in an arena completely filled with static water (i.e., without flow; $n = 11$ larvae, 3 – 4 larvae per assay in 3 experiments) (**revised Figure 4**). These experiments served as control to assess the effects of shear stress exposure due to continuous flow on the baseline swim behavior of the zebrafish larvae. Compared with control groups with flow, larvae perform more forward swims and less turns, while the overall swim frequency appeared similar (**Figure 4g–i**). The results demonstrated that the larval subjects do perform rheotaxis in our new flow-based behavioral assays. Nevertheless, such an effect has been accounted for and would not affect the interpretation of chemosensory-guided behaviors studied using our behavioral setup, since all other comparisons were made under identical conditions with continuous fluid flow (**Figure 4**). For the whole-brain imaging μ fluidic device, we additionally explained that our flow delivery design prevents co-exposure to new onset shear stress (**page 6, paragraph 2**), which is otherwise unavoidable with other conventional delivery methods.
- In the revised Discussion, we acknowledged the limitation that given the high sensitivity of odorant receptors, even small contralateral spillovers could elicit odor-induced activities in the olfactory epithelium unintended for stimulation and confound the experiments, and discussed the implications on the input selectivity and summation linearity results (**page 13, paragraph 4 to page 14, paragraph 1**). However, we also carried out additional analysis of unilateral olfactory stimulation to demonstrate the high lateral specificity of our chemical cue delivery (**Rebuttal Figure 1**), and consolidate the validity of our approach. We also discussed the value of adopting homogeneous odor landscapes for studying chemosensation enabled by our platform, despite the difference from the more complex, textured odor landscapes that an animal would encounter in nature (**page 14, paragraph 2**).

Please see below the detailed point-by-point responses to address each of the concerns below, supplemented with **Rebuttal Figure 1**.

Reviewer #1

Sy et al. present an optofluidic platform for studying zebrafish larvae behavior in the presence of chemical cues. The tools include precisely controlled chemical delivery, a microfluidic arena and an integrated light sheet microscopy system. Overall, the authors present an impressive setup with convincing data. The text is very well written.

General Response:

We are very grateful to the reviewer for the positive remarks on our manuscript.

Major remarks:

I find it a bit confusing how many different implementations of the microfluidic platform the authors have developed and how they are being used. Is the μ fluidic swimming arena in Fig. 1 actually used for experiments or is this rather a preliminary

test on the way to the more complicated setups later on? Fig. 2 and 3 show a very similar setup and it is again not clear if this represents two distinct setups or if these are just two different states during the development of the final system. All in all, this redundancy in the figures should be removed if possible or the authors need to explain in the text the sequence and importance of the different setups.

Along similar lines, the text is rather lengthy and would benefit from some shortening, making sure it becomes clear what has been why, step-by-step.

Response:

We thank the reviewer for the important comments regarding our presentation. We have revised **Figures 1–3** and added additional figure references at appropriate places in the main text to indicate the setups used for experiments. For the μ fluidic arena, we now indicate clearly that (**page 4, paragraph 2**): “*We thus verified that the μ fluidic arena (**Figure 1a**) can maintain a constant chemical landscape over time, and used it for chemosensory-guided behavioral assays (see later sections and data presented in **Figure 4**).” For the optofluidic setup for integrated stimulus delivery and functional imaging presented, we had revised **Figures 2 & 3** and their legends, and the main text (**page 5, paragraph 2**), by revising the corresponding sentences and using the words “*final*” or “*successful*” at appropriate places to indicate the final functional design (shown in **Figure 2b, lower panel**, and in **Figure 2e**), and to distinguish them from the alternatives which did not work (shown in **Figure 2b, upper panel**, and in **Figure 2d**). The schematics in the revised figures no longer present redundant information, but rather with one focusing on the μ fluidic design (**Figure 2**) and the other on the optics and system integration (**Figure 3**).*

We also made revisions to make the explanations on the design principles of the optofluidic system easier for readers to follow, by outlining the goals the optofluidic device first (**page 4, paragraph 3 to page 5, paragraph 1**): “*Such a device needs to accomplish several goals, including (i) stabilization of the larval subject, (ii) compatibility with neuronal imaging techniques, and (iii) precise control over chemical stimulus delivery.*” We then start each following paragraph with streamlined sentences specifying what we tackled next, for example in **page 5, paragraph 2** we shortened the first two sentences as: “*We first devised a fluidic channel design for the stabilization of a larval subject’s head in the micrometer range for high-quality neuronal activity imaging (**Figure 2a**).*”, whereas in the next ones we use these shortened sentences to make it clear our goals (**page 5, paragraph 3 to page 6, paragraph 1**): “*Next we ensured that the μ fluidic chip is compatible with volumetric neuronal imaging techniques.*”, “*We then tackled the need for accurate odor stimulation of individual or both olfactory placodes (OPs).*”, and “*We also optimized fluid stream control to improve the temporal precision of odor stimulation.*”.

The figures also show some information that is very hard to interpret. For example, in Fig. 1 images in panels c and e look identical. Maybe the contrast can be improved. The photos in Fig. 2d are hard to interpret. It is not clear that the glass window offers any advantage. It is not clear to me what Fig 2g shows. Is it the ratio or what does “vs” mean? Not sure if this is the best way to show these data.

Response:

Thank you for the important remarks. As advised, we had increased the contrasts of the images in **Figure 1c, e**, and revised the legends accordingly. For **Figure 2c** (original Figure 2d), we intended to show that with our customized μ fluidic fabrication technique, we obtained a flat and transparent sidewall that would not disrupt excitation light propagation. To ensure this point is clearly conveyed, we have added schematics to illustrate the difference.

In **Figure 2e, right panel** (original Figure 2g), we meant to present that it is the ratio that matters (but not L vs R). To avoid confusing the readers, we have revised the plot by merging the data points for the two intended sides (using different shapes to indicate the intended side of stimulus), while still marking the dotted line to show that the ratio of intended:unintended side was consistently > 20 as the emphasis.

Figure 3a is confusing as the geometry is not accurate (partially 2D/3D). This needs to be drawn in the correct geometrical perspective.

Response:

We have revised **Figure 3a** as advised to remove redundancy with the previous figure, and ensure a clearer 2D-only presentation for the optics.

Reviewer #2 (Remarks to the Author):

The authors have reported the application of two microfluidic devices, one for behavioral screening and one for light sheet brain-wide neuronal GCaMP6f imaging of zebrafish larvae. The former device allows the larvae to move freely in response to cadaverine brought into a chamber at its corners. The latter immobilize the head of the larva while allowing the tail to move in a chamber. The two nasal cavities of the immobilized larva are claimed to be exposed independently to chemicals using laminar flow in the microchannel. The manuscript is well written, and the results are substantial and well presented. There is also a good discussion on the results. I think the manuscript introduction requires substantial revision, and some more experiments are needed to back up some of the claims. If the authors can address these comments, I think the paper becomes suitable for publication in Nature Communications and useful for a wide range of researchers using zebrafish as a model for chemical screening.

General Response:

We are very grateful to the reviewer for the positive remarks that our results are substantial and well-presented, along with a good discussion. As advised, we have substantially revised the introduction and performed additional experiments to back up some of the claims, according to the reviewer's suggestions. Please see below for the detailed point-by-point responses.

Main Critique

One of the most important issues associated with the introduction section of the paper is the lack of a clear research hypothesis, followed by offering no research questions that are investigated. The introduction section is written mostly based on the technologies available to screen zebrafish. On the technology side, zebrafish larvae chemotaxis in chambers and the movement of semi-mobile larvae in microchips have both been demonstrated. LSM imaging is also well established in the literature. I felt that the prior literature in these areas was not cited thoroughly. The spatial resolution

in exposing the nasal cavities independently would be considered a marginal technology improvement. So overall, although very interesting in terms of technology integration and application., the work is not technologically novel enough for the esteemed Nature Communication. From the science perspective, as mentioned above, I failed to see a coherent introduction section with a clear hypothesis and research questions. They are offered in the results section though, and very interesting knowledge gaps have been addressed. So, I recommend re-writing the introduction based on the science investigated.

Response:

We thank the reviewer for the constructive critique. As advised, we have revised the introduction to provide a more thorough coverage of the background literature (**page 2, paragraph 2, with refs 1, 18–35; and page 3, paragraph 1, with refs 44, 47, 48, 53–56**). We further elaborated on the justification and strengths of our developed method with its unique spatial accuracy for unilateral olfactory stimulation (**page 2, paragraphs 3**): *“In sensory response mapping experiments, in order to dissociate the functional effects of activating individual vs. paired sensory organs, or ipsilateral vs. commissural pathways, a common practice is to selectively stimulate one or both sensory input channel(s). This allows experimenters to reveal how bilateral sensory organ inputs are represented, and the crosstalks between commissural pathways.”*

To provide a clear hypothesis and research questions that we intended to address as a proof-of-principle, we added the following passage (**page 3, paragraph 2**): *“We reasoned that precisely controlled odor stimulus profiles, integrated with behavioral and neuronal imaging, permit unbiased assessment of odorant-evoked behaviors (e.g., regarding valence, behavioral algorithms in pursuit or avoidance), and provide the necessary constraints for constructing neural circuit models among possible alternatives (Supplementary Figure 1). We hypothesized that this would allow us to address how larval zebrafish avoid potentially noxious chemicals, such as cadaverine – an odor produced by the putrefaction of corpses, and whether or how such avoidance depends on binasal inputs. In particular, we will address whether comparisons between the inputs from two sides are used to direct turns and whether swimming frequency and angular velocity are modulated to assist escape. Coupled with functional imaging, we may also uncover the underlying neural basis at the sensory representation level.”*

We would also like to point out the unique difficulties we have overcome, and advancements made over past reported systems (e.g., that reported in refs 47 & 48), to achieve the optofluidic design. Specifically, in our system a larval subject is (i) fully immersed in continuous fluid medium, and (ii) allowed to flip its tail during stimulus presentation, and that (iii) such movements must not compromise the quality of the cellular resolution imaging, or the single-olfactory placode spatial accuracy of chemical stimulus. These represent a unique combination of properties not accomplished by previous designs, and were made possible only with our innovations in μ fluidic design, including a carefully designed head and waist trap (**Figure 2a**), the incorporation of a side channel to dissipate pressure changes (**Figure 2b, lower panel**), customized fabrication to obtain a flat sidewall

and integration with light sheet microscopy (**Figure 2c & Figure 3a**), and the converging fluid streams design for stimulus delivery (**Figure 2e, f**).

In this revision, we have provided more evidence to support our claims on the stability of imaging during stimulus delivery and tail movements (**Supplementary Figures 2 & 3**). On the other hand, in the behavioral assay, we bridged μ fluidics and behavioral assays, and demonstrated the importance of precisely knowing the spatial chemical profile. In contrast to short-time scale puffing, a longer term assay (e.g. 2 hours) allowed the revelation of previously unknown valence and behavioral algorithms upon cadaverine encounter by larval zebrafish (also discussed in **page 12, paragraph 4 to page 13, paragraph 1**). We hence believe that our methods would greatly benefit the study of chemosensation, and thank the reviewer's positive remark that very interesting knowledge gaps have been addressed.

Major comments:

- 1. Exposure of the larva's head to the chemical stimulus is accompanied by co-exposure to mechanical shear stress. This shearing flow may also be present in the swimming arena device as there is a continuous flow into and out of the chamber, although being very small. Control experiments measuring the effect of these shears and separating them from chemical sensing phenomena observed in the two devices would be good to elaborate on. Did the authors attempt to minimize these flow rates to minimize the shearing effects?**

Response:

We thank the reviewer for the important reminder on the potential effects of shear stress in the two devices, which we address below:

For the behavioral arena: The flowing streams offer the advantage of allowing the experimenter to maintain a static chemical landscape, and hence unambiguous analysis based on knowledge of whether at any spatial location and time point, a larva under the assay is exposed to a chemical of interest. We however completely agree that the continuous flows in behavioral assays likely co-stimulate the lateral lines (as they are immersed in the fluid streams and exposed to shear stress) and elicit rheotaxis. To assay the effects of rheotaxis, we performed additional assays with larvae in the same arena without flow, i.e. filled by static water. As shown in the revised **Results** section (**page 9, paragraph 1**) and corresponding figure panels in **Figure 4**, the presence of flow principally induced a general increase in turn angle distribution (**Figure 4g**) and had a minimal effect on swim bout frequency (**Figure 4i**), compared to normal larvae in static arena control. This however did not affect the chemosensory behavioral assays testing the behavioral effects of cadaverine, or the assays regarding their dependence on the integrity of binasal inputs, since all experimental groups were compared under the same fair conditions with flow (see other panels in **Figure 4** showing the differences across the different experimental groups).

For the optofluidic system for functional imaging: We wish to clarify that with the sandwich flow scheme we adopted for biological experiments (**Figure 2f**), we maintained a constant flow during the baseline (i.e., before stimulus delivery) and post-stimulation. In other words, throughout each imaging trial, a constant shear stress profile in the vicinity of

each nasal cavity was maintained. The only difference is a general small decrease in the flow velocity magnitude, as the overall flow decreased with the cessation of the insulating water stream and the stimulus stream for stimulus delivery and removal, respectively (**Figure 2f**). This is in stark contrast to more conventional methods of delivery (e.g., via “puffs”, or forcible ejection of a fluid stream), which must involve the initiation of a new fluid stream and therefore shear stress exposure upon chemical cue delivery. Our stimulus method prevents such shear stress change (or co-exposure), and mimics that in nature, animals frequently swim across constantly flowing fluid streams to sample chemical cues. Importantly, the sensory organs processing mechanical cues are located at the lateral lines, which are not exposed to any shear stress in our device. We have revised the main text to make sure these points are more clearly conveyed and prevent misunderstandings (**page 6, paragraph 2**).

Regarding the selection of flow rates in the two devices: For the behavioral arena (**Figure 1a**), our principle was to attain a balance between effects on minimizing the effects of diffusion across fluid streams (which requires a higher flow rate), vs preventing turbulent mixing and reducing eliciting rheotaxis (both of which requires a lower flow rate). Both the Reynolds number (Re) and the Péclet number (Pe) are proportional to the flow rate. Given the chosen flow rate ($660 \mu\text{L min}^{-1}$, giving rise to a complete turnover of fluid in the arena only over every 4 mins), Re was > 3 orders of magnitude (0.427 , see details of calculation in **Methods, page 16, paragraph 1**) lower than the threshold for turbulent flow (i.e., 2000), indicating a very small flow that nonetheless could prevent turbulent mixing and avoid strong rheotaxis (as evident by that the larvae only exhibited more turning but without significant increase in swim bout frequency in the arena with flow, see **Figure 4i**). $Pe = 5.42 \times 10^2$ was also > 2 orders of magnitude higher than the threshold necessary for preventing significant diffusive mixing (i.e., 1). The flow rate was therefore on the low end of order of magnitude to ensure a high confidence on the chemical zone border, with a ± 1 mm-wide border (**Figure 1c, d**) – a width we consider appropriate for the length scales of larval zebrafish body length and their swim bouts (both in mm-scale). For the optofluidic device for whole-brain imaging (**Figure 2a**), note that the lateral lines of the larval subject are not exposed to flow changes. We therefore mainly had to balance the precision of chemical cue stimulus, as rise/decay time decreases with an increase in flow rate, while we also wish to prevent too high a baseline shear stress. We selected a flow rate just enough to allow for around 1-second stimulus delivery precision (**Figure 2f; page 6, paragraph 1**), appropriate for the time scale of olfactory perception.

2. **On page 4, there are claims that adding the side channel stabilizes the pressure that helps with minimizing the larva wobbling. These claims require experimental proof.**

Response:

As advised, we have added additional data to support this claim, by providing data from an example larva imaged in an μ fluidic device without side channel that wobbled with stimulus delivery (**Supplementary Figure 2f**), compared to our final design with the side channel that enabled stable imaging (**Supplementary Figure 2a–e**, also see **Supplementary 3**).

- 3. On page 5, the authors claim that the reverse flow is better than a forward flow for nasal cavity exposure. The provided experimental results are not sufficient to back up this claim. A very small spatial imprecision in larva positioning will hydrodynamically change the flow distribution profile regardless of the flow direction. Small larva movement may also affect the fluid distribution around the nasal cavities that was not discussed in the paper.**

Response:

We thank the reviewer for the important comment. We revised the paper to provide more examples comparing the results obtained with flow delivery in the two configurations (**Figure 2d, e**). Based on our observation, delivery from the front with diverging flows rarely allowed us to accomplish flow segregation, instead fluid stream spillover to the contralateral side typically occurs (6 example trials shown in new **Figure 2d**). In contrast, with reversed flow converging at the midline, we could reliably attain stream segregation without crossover (6 example trials shown in new **Figure 2e, left and middle panels**). In the test trials (**Figure 2e, right panel**) and biological experiments we almost always had a ratio of stimulus specificity quantified by 1 μ M sodium fluorescein fluorescence to be $> 20:1$ (intended vs unintended stimulus side, and in data analysis we only included such trials, see **Methods on page 21, paragraph 3**). Partly, this was accomplished based on the tailor-made trapping chamber that provides mechanical constraint for positioning (**Figure 2a**) and stabilizes the head (and therefore the nostrils) during tail flipping and stimulus delivery (also see **Supplementary Figure 3**).

- 4. What is the effect of diffusion on the distribution of the chemical at the nose location, potentially causing the chemical to diffuse from the target nasal cavity to the other cavity? This is important to report as the flow velocity is claimed to be very small. Fluorescent images of a dye molecule are not sufficient, as the dye is different from the chemical and the imaging sensitivity may not pick up the diffused dye if fluorescence is under the threshold. The authors can do some analytical calculations with the diffusion coefficient, in addition to the calculation of Peclet and Reynolds numbers.**

Response:

As advised, we have additionally estimated the effect of diffusion with Péclet numbers (in addition to the effect of turbulent mixing with Reynolds number), and found that the diffusion rate is 2 to 3 orders of magnitude lower than that of advection in both the behavioral and imaging setups, respectively (see **page 4, paragraph 1 & page 5, paragraph 4**, with detailed calculation in the **Methods** section on **page 16, paragraph 1 & page 18, paragraph 1**). We are grateful that this suggestion has helped us consolidate our claim.

- 5. LSFM imaging is very interesting in the presented paper. Where the LSFM images in Fig 3b acquired when the tail was moving? How did the movement affect the LSFM images?**

Response:

We thank the reviewer for commenting that the LSFM imaging is interesting. Indeed, although PDMS μ fluidic devices offer superior fluid control for stimulus delivery, stable imaging with a larval subject immersed in fluid medium in such devices is a challenging task in comparison to agarose-embedding. The LSFM images shown in **Figure 3a, bottom panel** (original Figure 3b) were acquired as an anatomical stack, where the exposure time for each plane was longer than functional imaging for higher signal-to-noise ratio and resolution for reference. During the acquisition of anatomical stacks, we did not monitor tail flipping but it was likely that the tail had flipped.

To better illustrate the effects of tail movement on imaging, we have additionally presented two sets of simultaneously acquired functional imaging and tail flipping time-series images in an example trial (**Supplementary Figures 2 and 3**). These serve to provide clearer illustrations of stable imaging during tail flipping and chemical cue delivery. We have also added the descriptions in the main text (**page 7, paragraph 1**). As described in the **Methods** section (**page 21, paragraph 2**), we used the well-established NoRMCorrE algorithm (developed by Pnevmatikakis et al., *J. Neurosci. Methods*, 2017; and used by Marquez-Legorreta et al., *Nat. Commun.*, 2022 and Shemesh et al., *Neuron*, 2020) to correct for micrometer-range drifts in functional imaging, and we only include trials without blurring of any imaging frames (i.e., due to larger movements of the head location during the acquisition of a given image) for further analysis.

6. I recommend adding a section and fully characterizing the specifications of the light sheet.

Response:

Thanks a lot for the important recommendation. We have now provided the light sheet and point spread function characterizations in the revised figure (**Figure 3a**) and main text (**page 6, paragraph 3**).

Minor comments:

1. Page 2, paragraph 2, line 2-3: The sentence reads as if no tools are available for precise chemical stimulation of small organisms. This is an incorrect message and should be revised to acknowledge the substantial work done in this area on C elegans, Drosophila and zebrafish using microfluidics.

Response:

We have revised the sentences and added suitable references (**page 2, paragraph 2, with refs 1, 18-35; page 3, paragraph 1, with refs 44, 47, 48, 53-56**). Please kindly advise us further should there be additional suitable references we should cite.

2. Please justify in the introduction section why it is important to expose nasal cavities to chemicals one at a time.

Response:

We are sorry for the unclear justification in the introduction. We have now revised the introduction as follows (**page 2, paragraph 3**): *“In sensory response mapping experiments, in order to dissociate dissociate the functional effects of activating individual vs. paired sensory organs, or ipsilateral vs. commissural pathways, a common practice is to selectively stimulate one or both sensory input channel(s). This allows experimenters to*

reveal how bilateral sensory organ inputs are represented, and the crosstalks between commissural pathways.”

3. **On page 5, the rise time average is much shorter than the decay time average in chemical stimulation. Can the authors discuss why? Also, how do this difference and large variation in stimulation time affect the experiments claimed to be highly controllable?**

Response:

The difference arises primarily from the difference in total flow rates in the two scenarios, whereby stimulus onset was accomplished by a change from 3 to 2 streams (i.e., the cessation of insulating water stream, see **Figure 2f, left panel**), while turning it off involved a change of 2 to 1 streams (i.e., the cessation of chemical stream, also see **Figure 2f, left panel**). In another word, it takes proportionally more time for 2 streams to refill the space originally occupied by 3, vs 1 stream to replace 2 (and hence the ~1 s rise time vs ~1.5 s decay time). The lateral convection of stimulus or water by gradual removal of a fluidic stream adopted by our method provides a smoother fluidic transition that is also under more precise monitoring than previous methods which involve initiation of fluidic streams and/or awaiting the cessation of stimulus by simple diffusion (e.g., Herrera et al., *Curr. Biol.*, 2021; Mattern et al., *Comm. Biol.*, 2020; Koide et al., *Cell Rep.*, 2018; Candelier et al., *Sci. Rep.*, 2015; Krishnan et al., *Curr. Biol.* 2014; Dreosti et al., *Curr. Biol.*, 2014). These points are now mentioned in the revised manuscript (**page 6, paragraphs 1 & 2**). Importantly, (1) these rise and decay times are in a time scale appropriate for olfactory perception, and (2) the calculation of mutual information has taken the temporal profile of chemical cue into account (see **Methods, page 21, paragraph 4 to page 22, paragraph 1**).

Reviewer #3 (Remarks to the Author):

The authors present a technically demanding and impressive study to analyse neural circuits activated by odors, together with the behavior elicited by these odors. Larval zebrafish are used as model system due to their transparency and small size, which allows whole brain imaging. Tail flipping movements can be recorded in parallel with imaging, unfettered movements are recorded in a separate arena with unusually high spatiotemporal control of odor stimuli (on a tens of μM scale). The main (and important) improvement beyond previous such studies concerns the precise control over odor concentration and extension of the odor plume in both settings. The accuracy of the imaging setup is sufficient to stimulate the left and right nostril of the zebrafish larvae separately, thus allowing to demonstrate interhemispheric connections.

The authors have used the aversive odor cadaverine to demonstrate the feasibility of their setup. Adult zebrafish were known to react with aversive behavior to cadaverine, but the response of larvae was not clear from previous experiments. Moreover the authors have imaged the brain networks activated by cadaverine with uni-nasal and bi-nasal odor stimulus, and have compared the behavior of larvae with one olfactory placode removed (to achieve uni-nasal input) with untreated control larvae. The evaluation of the data appears to be thorough.

Reported biological results are:

- 1. Zebrafish larvae do strongly avoid cadaverine (but see below)**
- 2. Larvae with only one olfactory placode need more movements to leave the cadaverine zone. This is interpreted as binasal input being required for proper motion control.**
- 3. Stimulus-induced activity occurs on the contralateral side already in the olfactory bulb, and all more posterior brain regions.**
- 4. Stimulus-induced activity for animals with binasal input is supralinear compared to stimulus presented only to one nose.**

Taken together this is a technically very strong paper, with novel findings enabled by the improved methods, but I have some doubts concerning parts of the biology (see below).

General response:

We are grateful to the reviewer for the encouraging remarks that this is a technically very strong paper enabled by our methods. We also thank the reviewer for acknowledging the methodological advancements and novel findings revealed. As advised, we have included more experiments, data and analyses to address the concerns raised regarding some parts of the biology.

Major points:

p6: The authors use 1 mM cadaverine for the avoidance assays and state that no immediate effect was seen, but over 2 hours there was strong avoidance. The authors also show that after 2 hours in 1 mM cadaverine the large majority of larvae is already dead. Thus it is unclear, whether the avoidance results from olfactory sensing of cadaverine – it could also come from sensing malaise (a standard method to strongly induce taste aversion is based on such sensing). This could be examined by bilateral ablation of the olfactory placodes. If the author's interpretation is correct, the avoidance behavior should no longer occur. Such control experiments should be included.

Response:

We thank the reviewer for the important remark that the suggested control experiments are important to examine the non-olfactory component(s) in cadaverine avoidance. As advised, we have now included the suggested experiments with bilateral ablation of the olfactory placodes (designated null-OP (nOP) larvae group) in the cadaverine avoidance assay (revised **Figure 4**). Comparing nOP larvae against other groups, we observed the following phenomena:

(i) nOP larvae could still avoid cadaverine, and to a similar extent as larval subjects with unilateral intact OP (uOP larvae). However, both nOP and uOP larvae did not avoid cadaverine as efficiently as larvae with bilateral intact OPs (i.e., normal larvae, designated bOP larvae) (**Figure 4a–e**).

(ii) Upon encountering cadaverine, the turn angle and angular velocity increased only with at least one intact OP, and in a *graded* manner (i.e., the bOP larvae made larger and faster turns than the uOP larvae) (**Figure 4g, h**).

(iii) In contrast, single OP could not drive swim bout frequency increase beyond that observed in nOP larvae (**Figure 4i**). bOP larvae made substantially more frequent swim bouts than both the uOP and nOP counterparts.

We thus made the following interpretation: In larval zebrafish, cadaverine avoidance indeed has a non-olfactory component (e.g., via gustation). Single OP input could only increase turn angle and velocity but not swim bout frequency. Although detecting cadaverine with single or both OPs appeared to drive the same behavioral algorithm, sensing cadaverine via both OPs is crucial to optimize the avoidance behavior with higher swim bout frequency and large-magnitude angular velocity increase (**page 10, paragraph 4**).

We believe while our data pointed to a non-olfactory component of cadaverine avoidance, it does not change or weaken the conclusion that the avoidance behavior is bilateral OP-dependent. It also does not affect the methodology as the main emphasis of the manuscript. Rather, these findings revealed the necessity of a systematic approach with precise control over chemical landscape (e.g. that permitted by our setups), to reveal the potentially complex chemosensory driven behaviors induced by different chemicals.

The authors seem to have used a single and rather high cadaverine concentration, i.e. 1 mM. Why was no lower concentration used? Ref 75 shows imaging results with 100 μ M cadaverine. Using 100 μ M could have avoided the problem with mortality (see above). Furthermore, with the 20:1 specificity for unilateral odor presentation reported by the authors (p.5), 1 mM stimulus still means 50 μ M on the contralateral side, which could be enough for a response. This problem would be much reduced with 100 μ M starting concentration.

Response:

We thank the reviewer for this important comment regarding the choice of concentration used, and we agree that even a small spillover could confound the interpretation of results. We now acknowledge this weakness in the revised Discussion (**page 13, paragraph 4 to page 14, paragraph 1**).

We wish to explain our original rationale of picking 1 mM cadaverine for assays: Firstly, We took reference to a prior study demonstrating that adult zebrafish avoid 1 mM cadaverine provided in puffs in a tank, with the responses were much weaker below 1 mM, and saturating at concentrations greater than 1 mM (ref 73: Hussain et al., *PNAS*, 2013). Since there were uncertainties regarding whether larval zebrafish can avoid cadaverine, we heuristically chose a 1 mM to ensure a better chance of the behavior revealed in the assays. Secondly, as both our behavioral and imaging results showed differential responses comparing unilateral vs bilateral olfactory cadaverine stimulation at 1 mM, we believe it represents a suitable concentration to demonstrate the capabilities of our behavioral and optofluidic setups.

For the behavioral assay, we wish to clarify that no larval subjects died in the avoidance assay over two hours, as the cadaverine zone was restricted only to the right side of the arena and the larvae had limited cadaverine exposures. In the avoidance assays, all larvae were able to escape the cadaverine zone in less than 45 minutes, even in the longest cadaverine zone entry-to-escape event. In fact, only 1 out of 569 entry-to-escape events was longer than 15

mins (with the singular event being 40.8 minutes), while the median escape times were < 0.7 seconds for all avoidance assay groups (i.e., including bOP, uOP, and nOP larvae in cadaverine avoidance assays) (**Figure 4e**) – a time scale substantially shorter than that would result in death (minimally 60 mins, see **Supplementary Figure 4a**). In the avoidance behavioral assays, the health of the larval subjects were therefore unlikely impaired. Mortality was only observed in the survival assay with the arena completely filled with 1 mM cadaverine (i.e., enforcing prolonged compulsory exposure) (**Supplementary Figure 4a**), with which we meant to demonstrate the ecological relevance of the avoidance behavior (**page 7, paragraph 3**).

For the imaging assays, we agree with the reviewer that zebrafish can indeed sense cadaverine over a large range of concentrations (e.g., even in the range of 1 – 10 μ M, *cf* Li et al., *eLife*, 2015), and that a lower concentration used for unilateral stimulation may further ensure lateral stimulus specificity. To mitigate this potential issue, in our analysis, we only included trials with at least > 20:1 ratio of intended vs unintended stimulus intensity. In addition to data shown in **Figure 5** and **Supplementary Figures 6 & 7**, we have re-analyzed the responses of more olfactory sensory neurons (OSNs) (**Rebuttal Figure 1**). In fact, for the large majority of the OSNs, we observed no stimulus-locked activation during contralateral olfactory stimulation (**Rebuttal Figure 1**). This further consolidated the confidence in the validity of our technical approach and data included for analysis. We also wish to point out that some results we reported are more “resilient” to the potential maximal 1/20 spillover. For example, it would not compromise the interpretations that there is a wide range of ipsilateral-contralateral nasal input selectivity across brain regions (**Figure 5c–e**), or the supralinear summation of binasal inputs (**Figure 5f–h**), as spillover would only tend to lead to an underestimation of these effects. We have additionally discussed these points in the revised manuscript (**page 13, paragraph 4 to page 14, paragraph 1**).

Overall, we thus believe that the use of 1 mM cadaverine did not compromise our study’s technological strengths or affect the main conclusions of the biological findings. We certainly agree that further experiments testing the effects of more concentrations of cadaverine (and other chemicals) would be required to get down to further details of chemosensation, however these are beyond the scope of our current study emphasizing the methodology development and proof-of-principle of its use.

p13: The authors state in Methods that they used 5-7 dpf zebrafish larvae. 5 dpf larvae presumably have not yet fed, but 6 and 7 dpf have taken up food. Can the authors exclude that feeding experience does influence the observed behaviors? How was the age distribution for the individual experiments?

Response:

We apologize that information regarding the feeding experience of the larvae was not included in the original manuscript. We have now provided this information in the **Methods** section (**page 15, paragraph 1**), that “*For all experiments reported in this study, we used 5–7 days post fertilization (d.p.f.) larvae carrying the nacre mutation. Larvae were raised and maintained under 14-h light:10-h dark cycles at 28°C. All larvae used for experiments were not fed (which would not compromise their health; cf ref 94).*”. When raising zebrafish larvae, feeding protocols adopted by different teams may vary on the start day for feeding.

Importantly, a prior study showed that delayed feeding to 8 d.p.f. has no impact on the health or survival (compared at 39 d.p.f.) of larvae (ref 94: Hernandez et al., *Zebrafish*, 2018). Please also kindly find the specific age distributions for the behavioral and imaging experiments in the **Methods** section in the manuscript (**page 16, paragraph 3; page 19, paragraph 3**).

p14: The larvae with unilateral ablation of olfactory placode (uOP) are treated rather differently from the control animals (bOP). I could not find any description of a mock procedure for the bOP, to attempt to control for the stress induced in the uOP (by extra embedding on 4-5 dpf and 2 photon ablation, plus additional 2 photon analysis before the experiment). Can the authors exclude that these animals were in generally poorer shape and therefore show hampered motion?

Response:

We thank the reviewer for the important comment regarding the effects of ablation procedures (i.e., extra embedding, imaging and nasal placode ablation) on the health status and swim behaviors. As also mentioned in the manuscript (**page 4, paragraph 1**), an advantage of the arena design we wish to emphasize is that the mirror zone in fact served as a convenient control for assessing baseline motor activities (including rheotaxis). Please refer to the mirror zone data (**Supplementary Figure 4c–i**), with which we showed similar baseline swim kinematic parameter distributions for larvae that had or had not undergone the olfactory placode ablation and two-photon imaging procedures. These control data showed that the ablation and imaging procedures had no adverse effects on the larvae's health, and did not hamper motion.

A general point, which could be touched upon in the discussion: The strength of the paper is the precise control and temporal and spatial homogeneity of the odor stimulus. However, this is also a fairly unnatural situation, as in nature the animal would navigate a 'textured' plume of odor.

Response:

We thank the reviewer for the kind suggestion. We have revised the last paragraph of the Discussion section (**page 14**): *“In nature, an animal would navigate more complex odor landscapes. Yet, the precise spatiotemporal control and homogeneity of odor stimulus in the optofluidic setup allow systematic analyses of the neural representation of individual odorants and evoked behaviors. The high-precision delivery system is also extensible to accommodate an elaborate array of chemicals varying in species, concentrations and valence, and uncover the corresponding sensory representations of odorants and their mixtures.”*, to better explain the value of the approach, despite experimentally it may not fully resemble the natural environment.

Minor comments:

p2, „As ecologically the oldest sensory systems¹⁷, our understanding of the array of chemosensory-guided behaviors and the neural basis of chemical senses remains incomplete.“

Sentence logic incorrect, please rephrase. Also I suspect 'evolutionary' is meant, not 'ecological'?

Response:

We have replaced “ecological” with “evolutionary” as advised (**page 2, paragraph 2**).
p3, „we show that the neural representation of cadaverine sensing is characterized by a wide range of ipsilateral-contralateral nasal input selectivity“

Unclear. What is a 'wide range of nasal input selectivity'?

Response:

We have now elaborated on the sentence to ensure a clearer description (**page 3, paragraph 2**).

p3, „the laminar fluid streams would predominantly shear across each layer,,

What does 'shear' mean in this context?

Response:

We have revised the phrase to “...*the laminar fluid streams would predominantly slide (or shear) across each layer...*” to make it more intuitive to understand (**page 4, paragraph 1**).

p3, „Computational fluid dynamics simulation showed laminar flow at steady state“

'Suggest' would be a better term than 'show', since it is a simulation.

Response:

We have replaced the word as suggested (**page 4, paragraph 1**).

p5, „(1.64 ± 1.04 s (mean ± S.D.)“

This should at most be given as '1.6 +/- 1.0', the second digit does not carry information.

Response:

We have dropped the second digit as advised (**page 6, paragraph 1**).

Figure 3B, p6 top

The authors claim „Donut-shaped neurons could be seen in most parts of the brain“

It would be preferable, if the authors point out some neurons by suitable symbols.

Response:

We have added some arrows to point out some neurons in the corresponding panel in revised **Figure 3a**.

It would also be helpful, if the authors could give a (rough) estimate about the frequency of labeled neurons, e.g. per total cells or per area.

Response:

We have now added statistics on the number of detected neurons (**page 21, paragraph 2**).

p6, „In 9 behaviorally active larvae, the distribution of spontaneous tail flipping frequency „

The age of these larvae is not given. Since the behavior of young larvae depends very much on their age, this is an important information, which should be given.

Response:

The age range (all 5 – 6 d.p.f. at time of experiments) of the larvae used for whole-brain neuronal imaging is now specified in this sentence in **Results (page 7, paragraph 2)**, in addition to **Methods (page 19, paragraph 3)**.

p8, „We did not find the larvae performing significant linear orthokinesis „

The paper would be easier to read for the non-specialist if these and other technical terms would be replaced by their explanations (given on p7) – i.e. 'linear velocity' for 'linear orthokinesis' and 'swim bout frequency' for 'klinokinesis'.

Response:

We have replaced the terms with their explanations (**page 9, paragraph 3 to page 10, paragraph 2; Figure 4f**).

p8, „We identified sensory-encoding neurons, defined as those with the most significant mutual information^{72,73} between activity and cadaverine stimulus „

Please explain 'mutual information'. How does it account for 'noisy baseline'?

Response:

We have revised the corresponding passage to ensure a better explanation (**page 11, paragraph 1**): *“We used mutual information since it is the most general quantitative measure of dependencies between variables. This also took into account the noisy nature of neuronal activity signals, since its calculation requires the estimation of the joint probability distribution between stimulus and neuronal activity (i.e., taking the signal variables as statistical in nature).”*

p9, „In our dataset, in most larvae the left Hb neurons were similarly activated as the right counterparts by cadaverine (at 1 mM).“

Maybe speculate a bit why this result is different as that obtained in ref. 75.

Response:

We have added the speculation that this may be in part due to the higher concentration of cadaverine we used, in the main text (**page 11, paragraph 1**).

p9, „there is an increasing trend of FIs (Figure 5g, h) as IS values decrease „

Do the authors mean that there is a trend of Fis to increase?

Response:

Yes, we have rephrased the sentence to make it clearer (**page 12, paragraph 1**): *“Interestingly, apart from observing a supralinear gain of information for the majority of neurons (Figure 5b, f–h), there is an trend for F_{IS} to increase along the rostral-caudal axis (Figure 5g, h) as I_S values decrease across the forebrain regions (Figure 5a).”*

p10, bottom, „reported lack of cadaverine avoidance by larval zebrafish^{29,63}.“

Ref. 63 is a 1982 paper about the alarm response of zebrafish. Cadaverine is not mentioned in this paper.

Response:

We thank the reviewer for pointing out the error. We have deleted the citation (**page 13, paragraph 1**).

p15, „Slight customizations were made to the larva head and waist-trapping chambers to allow the fitting of either a right-tilted or upright oriented, right eye-ablated larva.“

Please explain the mechanical stress (or lack thereof) for the larvae in these chambers.

Response:

As advised, we have added the explanation in the revised main text (**page 19, paragraph 3**): *“To minimize mechanical stress in the chamber that the larvae may experience, we only picked suitably sized larvae that fitted the chamber volume well for the whole-brain imaging experiments.”*

Larval subject

Rebuttal Figure 1. More example trial-averaged responses to ipsilateral (Ipsi-STIM, orange), contralateral (Contra-STIM, violet) or bilateral (b-STIM, cherry) olfactory stimulation of individual regions-of-interest (ROIs) corresponding to cadaverine-responsive olfactory sensory neurons (OSNs) in the olfactory epithelia (OE) from four imaged larval zebrafish. For larval subjects *f315*, *f321* and *f418*, 15 OSNs were randomly sampled and shown, whereas *f334* had only 11 cadaverine-responsive OSNs detected. Shadow shows SEM across trials for each trace. Dashed rectangle indicates stimulus window. Scale bars: 10 seconds (horizontal) and 0.5 normalized dF/F (vertical).

REVIEWER COMMENTS

Reviewer #4 (Remarks to the Author):

The authors reported an optofluidic tool (termed μ fluidics-LSFM) to study chemosensory behavior and whole-brain neuronal activities. The microdevice allowed for controlled chemical delivery and behavioral imaging at cellular resolution. Using the microsystem, they revealed the neural representation of cadaverine sensing. The μ fluidics-LSFM also demonstrated its potential for studying chemosensory behaviors in other organisms, such as *C. elegans* and *Drosophila* larvae. Overall, the manuscript written well, and is important for the following exploration in brain neuronal activities with an advanced tool.

Based on the response, I think that the authors have addressed the most of the comments raised by reviewer #2.

In addition, in figure 1, the swimming arena represents a mesoscale and even macroscale device with 3D fluidic mixing, but not microscale (microfluidics) 2D mixing. The authors are required to provide accurate presentation.

Reviewer #5 (Remarks to the Author):

Reviewer 3 raised important methodological concerns regarding the use of a highly concentrated (1 mM) stimulus, which is well above the threshold for most olfactory receptors, including that detecting cadaverine. This increases the possibility that other sensory modalities contribute to the avoidance to cadaverine (in addition to taste and chemically-induced malaise, pH sensing might play a role), as well as the potential for spillover to the contralateral side.

In their revised manuscript, the authors have performed additional experiments, new data analysis and edited the text to address these concerns. Overall, the revised manuscript provides a satisfactory answer to the concerns raised as explained in more details below

Please also see additional request for:

- clarification of missing significance level in Fig 4d
- edit text where cadaverine is described as an 'alarm' cue; cadaverine is aversive and avoided by zebrafish but fails to elicit defensive behaviors (e.g. freezing) induced by alarm odorants like schreckstoff, chondroitin sulfate, etc (see for example refs 73,90).

As recommended by Reviewer 3, the authors now include an additional experimental group, with bilateral ablation of the olfactory placodes (nOP) to control for putative non-olfactory component in cadaverine avoidance. They find that larvae with bilateral ablation of the olfactory placodes display reduced avoidance of the cadaverine zone over a period of 2 hours compared to intact larvae (Fig 4a, 2% time spent in cadaverine zone in intact larvae vs 14% in nOP larvae), but display similar avoidance levels as unilaterally ablated OP fish (Fig4a, 14% spent in cadaverine for both uOP and nOP). This indicates that cadaverine avoidance is mediated by olfaction in conjunction with other unspecified sensory modalities, which is not surprising given the high stimulus concentration used. When reporting the results of Fig4 in the text, the authors state that "[...] the bOP larvae on average traveled significantly shorter distances (Figure 4d) [...] to leave the cadaverine zone than the uOP and nOP larvae." However, the statistical comparison between bOP and nOP larvae is missing from Fig 4d. Please add the p value if it has been omitted, or edit the text accordingly to reflect marginal/lack of significance.

When comparing the behavioral reaction of the different experimental groups during escape from the cadaverine zone, the authors find that only larvae with intact olfaction increase escape bout frequency, and that the high average turn angles and turn angle velocity measured in intact larvae decrease in a graded fashion in unilateral and bilaterally ablated fish (Fig4 g & j). This supports the role of olfaction in rapid and efficient escape strategies to avoid cadaverine. Overall, together with these additional experiments, the data convincingly demonstrate that intact bilateral olfactory inputs are necessary to preserve the magnitude and efficiency of cadaverine avoidance, and provide a more nuanced picture of the sensory mechanisms underlying it.

Regarding the use of high concentration of cadaverine and how residual cadaverine spillover to the contralateral side might activate OSNs, Rebuttal figure 1 illustrating the responses of OSN to contralateral stimulation should be included in the manuscript. Note that active sampling of the water surrounding the nostrils mediated by beating cilia has been reported in fish (<https://doi.org/10.1016/j.cub.2016.11.036> ; doi:10.1098/rsif.2007.1281) and might contribute to the spillover.

Regarding the concern raised on the lack of mock procedure in controls compared to placode-ablated larvae, the data presented in Sup Fig 4 (now updated to include bilaterally ablated larvae) provides a reasonable demonstration that larval locomotion is essentially unaffected by the ablation procedure.

Rebuttal Letter

We thank both reviewers for the positive evaluations and encouraging remarks, as well as for the remaining advice, based on which we have further revised the manuscript. Please see below the point-by-point responses.

Reviewer #4

The authors reported an optofluidic tool (termed μ fluidics-LSFM) to study chemosensory behavior and whole-brain neuronal activities. The microdevice allowed for controlled chemical delivery and behavioral imaging at cellular resolution. Using the microsystem, they revealed the neural representation of cadaverine sensing. The μ fluidics-LSFM also demonstrated its potential for studying chemosensory behaviors in other organisms, such as *C. elegans* and *Drosophila* larvae. Overall, the manuscript written well, and is important for the following exploration in brain neuronal activities with an advanced tool.

Based on the response, I think that the authors have addressed the most of the comments raised by reviewer #2.

General response:

We are very grateful for the reviewer's encouraging remark that our study is important for the following exploration in brain neuronal activities with an advanced tool, and endorsing that we have addressed the comments raised by reviewer #2.

In addition, in figure 1, the swimming arena represents a mesoscale and even macroscale device with 3D fluidic mixing, but not microscale (microfluidics) 2D mixing. The authors are required to provide accurate presentation.

Response:

We thank the reviewer for this important reminder. We have revised the abstract (**page 1**), main text (**page 3, paragraphs 1, 3 & 4; page 7, paragraphs 2 & 3; page 8, paragraph 2; page 13, paragraph 1; page 14, paragraph 2**), **Methods (page 15, paragraph 2)** and **Figure 1 legends** to more accurately refer to the behavioral arena as a meso- to macro-scale, fluidics-based swimming or navigation arena. We also additionally clarify in the main text that while most of the flows in the behavioral arena are restricted to 2D (in between the PMMA layers separated just by 1.5 mm), and yet strictly speaking there could indeed be a very small amount of 3D fluidic mixing (**page 4, paragraph 1**), to ensure the most accurate presentations.

Reviewer #5

Reviewer 3 raised important methodological concerns regarding the use of a highly concentrated (1 mM) stimulus, which is well above the threshold for most olfactory receptors, including that detecting cadaverine. This increases the possibility

that other sensory modalities contribute to the avoidance to cadaverine (in addition to taste and chemically-induced malaise, pH sensing might play a role), as well as the potential for spillover to the contralateral side.

In their revised manuscript, the authors have performed additional experiments, new data analysis and edited the text to address these concerns. Overall, the revised manuscript provides a satisfactory answer to the concerns raised as explained in more details below.

Please also see additional request for:

- clarification of missing significance level in Fig 4d
- edit text where cadaverine is described as an 'alarm' cue; cadaverine is aversive and avoided by zebrafish but fails to elicit defensive behaviors (e.g. freezing) induced by alarm odorants like schreckstoff, chondroitin sulfate, etc (see for example refs 73,90).

General response:

We are very grateful for the reviewer's positive remarks and endorsing that we had provided a satisfactory answer to concerns raised by reviewer #3 in the previous round. We have updated the missing labeling or description for significance level in **Figure 4d** and the legends (kindly see below for details), and edited the manuscript to refer to cadaverine as an aversive cue (instead of alarm cue) as advised (**page 7, paragraph 3**).

As recommended by Reviewer 3, the authors now include an additional experimental group, with bilateral ablation of the olfactory placodes (nOP) to control for putative non-olfactory component in cadaverine avoidance. They find that larvae with bilateral ablation of the olfactory placodes display reduced avoidance of the cadaverine zone over a period of 2 hours compared to intact larvae (Fig 4a, 2% time spent in cadaverine zone in intact larvae vs 14% in nOP larvae), but display similar avoidance levels as unilaterally ablated OP fish (Fig4a, 14% spent in cadaverine for both uOP and nOP). This indicates that cadaverine avoidance is mediated by olfaction in conjunction with other unspecified sensory modalities, which is not surprising given the high stimulus concentration used.

When reporting the results of Fig4 in the text, the authors state that “[...] the bOP larvae on average traveled significantly shorter distances (Figure 4d) [...] to leave the cadaverine zone than the uOP and nOP larvae.” However, the statistical comparison between bOP and nOP larvae is missing from Fig 4d. Please add the p value if it has been omitted, or edit the text accordingly to reflect marginal/lack of significance.

When comparing the behavioral reaction of the different experimental groups during escape from the cadaverine zone, the authors find that only larvae with intact olfaction increase escape bout frequency, and that the high average turn angles and turn angle velocity measured in intact larvae decrease in a graded fashion in unilateral and

bilaterally ablated fish (Fig4 g & j). This supports the role of olfaction in rapid and efficient escape strategies to avoid cadaverine. Overall, together with these additional experiments, the data convincingly demonstrate that intact bilateral olfactory inputs are necessary to preserve the magnitude and efficiency of cadaverine avoidance, and provide a more nuanced picture of the sensory mechanisms underlying it.

Response:

We thank the reviewer for the detailed evaluation and positive comment that our data convincingly demonstrated that intact bilateral olfactory inputs are necessary to preserve the magnitude and efficiency of cadaverine avoidance. As advised, we have included the marginally significant *P*-values (i.e., near but above 0.05), in **Figure 4d** and **Supplementary Figure 5b**, and specified in **Figure 4** and **Supplementary Figures 4 and 5 legends** that all other *P*-values not marked across groups in the sub-panels were due to lack of statistical significance. We have also removed the word “significantly” (**page 8, paragraph 4**) to more accurately report all the results, including the marginally significant ones.

Regarding the use of high concentration of cadaverine and how residual cadaverine spillover to the contralateral side might activate OSNs, Rebuttal figure 1 illustrating the responses of OSN to contralateral stimulation should be included in the manuscript. Note that active sampling of the water surrounding the nostrils mediated by beating cilia has been reported in fish (<https://doi.org/10.1016/j.cub.2016.11.036> ; [doi:10.1098/rsif.2007.1281](https://doi.org/10.1098/rsif.2007.1281)) and might contribute to the spillover.

Response:

We thank the reviewer for this very insightful advice. We have now included the contents presented in the original Rebuttal Figure 1 now as **new Supplementary Figure 7**. In the revised manuscript, we have additionally cited the study and mentioned active sampling via beating cilia as a potential source of spillover (**page 14, paragraph 2**).

Regarding the concern raised on the lack of mock procedure in controls compared to placode-ablated larvae, the data presented in Sup Fig 4 (now updated to include bilaterally ablated larvae) provides a reasonable demonstration that larval locomotion is essentially unaffected by the ablation procedure.

Response:

Thank you again for endorsing that we have provided a reasonable demonstration for the previous concern raised.